# Assay to visualize specific protein oxidation reveals spatio-temporal regulation of SHP2

Ryouhei Tsutsumi [1], Jana Harizanova[2], Rabea Stockert[2], Katrin Schröder [3], Philippe I.H. Bastiaens[2,4] & Benjamin G. Neel[1]

Reactive oxygen species are produced transiently in response to cell stimuli, and function as second messengers that oxidize target proteins. Protein-tyrosine phosphatases are important reactive oxygen species targets, whose oxidation results in rapid, reversible, catalytic inactivation. Despite increasing evidence for the importance of protein-tyrosine phosphatase oxidation in signal transduction, the cell biological details of reactive oxygen species-catalyzed protein-tyrosine phosphatase inactivation have remained largely unclear, due to our inability to visualize protein-tyrosine phosphatase oxidation in cells. By combining proximity ligation assay with chemical labeling of cysteine residues in the sulfenic acid state, we visualize oxidized Src homology 2 domain-containing protein-tyrosine phosphatase 2 (SHP2). We find that platelet-derived growth factor evokes transient oxidation on or close to RAB5+/ early endosome antigen 1– endosomes. SHP2 oxidation requires NADPH oxidases (NOXs), and oxidized SHP2 co-localizes with platelet-derived growth factor receptor and NOX1/4. Our data demonstrate spatially and temporally limited protein oxidation within cells, and suggest that platelet-derived growth factor-dependent "redoxosomes," contribute to proper signal transduction.

[1] Laura and Isaac Perlmutter Cancer Center, NYU Langone Health, 430 East 29th Street, New York, NY 10016, USA. [2] Department of Systemic Cell Biology, Max Planck Institute of Molecular Physiology, Otto-Hahn-Street 11, 44227 Dortmund, Germany. [3] Institute of Physiology, Goethe-University, Theodor-Stern-Kai 7, 60590 Frankfurt, Germany. [4] Faculty of Chemistry and Chemical Biology, Technical University of Dortmund, Otto-Hahn Street 11, 44227 Dortmund, Germany. Correspondence and requests for materials should be addressed to R.T. (email: Ryohei.Tsutsumi@nyumc.org)

Multiple studies suggest that reactive oxygen species (ROS) (e.g., superoxide ($O_2^-$), hydrogen peroxide ($H_2O_2$), nitric oxide (NO)) are not merely toxic byproducts of cellular metabolism, but also function as second messengers that regulate specific signaling molecules[1]. Various stimuli, including cytokines and growth factors, such as interleukin-1 (IL-1), tumor necrosis factor-α (TNFα) and platelet-derived growth factor (PDGF), transiently evoke ROS production, and receptor-evoked ROS are required for precise regulation of at least some signal transduction events[1]. ROS can damage cellular macromolecules, suggesting that signal transduction-associated ROS must be regulated in a spatio-temporal manner. Several reports argue that production of ROS in response to IL-1 or TNFα occurs in a specialized endosomal compartment, which has been termed the "redoxosome"[2]. Whether redoxosomes contribute to other types of signaling pathways (e.g., by classical growth factors) has remained unclear, and the identity of specific proteins oxidized by redoxosomes has remained elusive.

Protein-tyrosine phosphatases (PTPs) regulate intracellular signal transduction by receptor tyrosine kinases (RTKs), cytokine receptors and integrins[3]. All PTPs share a conserved active site "signature motif," -[I/V]HCSXGXGR[S/T]G-, featuring an unusually acidic catalytic cysteinyl (Cys) residue that executes a nucleophilic attack on substrate phosphotyrosyl (p-Tyr) residues[4]. The same properties that confer a low $pK_a$ on the catalytic cysteine also render it highly susceptible to oxidation[3–5]. Consequently, PTPs have emerged as important ROS targets, which undergo transient oxidation and inactivation downstream of various upstream stimuli[5–7].

In response to physiological levels of ROS, PTP catalytic Cys residues are oxidized to the sulfenic acid state (SOH). Depending upon the specific enzyme, this Cys-SOH rapidly reacts with the adjacent main chain amido-nitrogen to form an intramolecular sulfenylamide (S–N) bond[7, 8], or with a vicinal cysteinyl residue to form an intra- or intermolecular disulfide (S–S) bond[7]. These oxidized states of PTPs are reversible, and can be reduced by the glutathione (GSH) or thioredoxin systems. Higher levels of ROS result in biologically irreversible PTP oxidation to the sulfinic, sulfonic, or sulfone states[7]. ROS-dependent, reversible inactivation of PTPs is believed to help fine tune phosphotyrosine-based signal transduction[1, 6, 7]. Support for this concept has been obtained mainly by biochemical approaches[9–12], as technical limitations have, in general, precluded investigation of the spatio-temporal nature of PTP oxidation.

SHP2, encoded by *PTPN11*, is required for normal RAS/mitogen-activated protein kinase activation by multiple growth factors, including PDGF, epidermal growth factor (EGF), and many others[13, 14]. SHP2 reportedly is oxidized in response to multiple cellular stimuli, including PDGF[15, 16], but the spatio-temporal dynamics, regulatory mechanism, physiological role and source of ROS for SHP2 oxidation have not been defined.

Here, we report an easy, sensitive, and potentially general method that specifically visualizes oxidized proteins. Focusing mainly on PDGF signaling, we find that SHP2 oxidation in response to cell stimulation is regulated spatially and temporally. Combining our approach with "nearest neighbor" object-based image analysis, we show that SHP2 oxidation occurs at, or in close proximity to, RAB5+/early endosome antigen 1 (EEA1)-endosomes, and requires specific cellular NOX proteins. Our results implicate "redoxosomes" in PDGF-dependent ROS production and signaling. Our assay also can visualize oxidation of another PTP, protein-tyrosine phosphatase 1B (PTP1B), in response to insulin.

## Results

### Visualization of oxidized SHP2 by dimedone-proximity ligation assay (PLA).
Dimedone is a cell-permeable, soft nucleophile that reportedly labels cysteine-sulfenic acids, but not cysteines in the thiol, disulfide, sulfinic or sulfonic acid state[17, 18]. To evaluate the applicability of these observations to PTPs, we assessed the specificity of dimedone in vitro using PTP1B, the oxidation profile of which was established previously[8]. As monitored by in vitro phosphatase activity, purified recombinant PTP1B sequentially underwent reversible, followed by irreversible, oxidation, in response to increasing concentrations of $H_2O_2$[8] (Supplementary Fig. 1a). Also, as expected[19], PTP1B was oxidized irreversibly by treatment with pervanadate (Supplementary Fig. 1a). We then labeled PTP1B with dimedone (5 mM) in the presence of dithiothreitol (DTT), $H_2O_2$, or pervanadate, and detected dimedone-labeled PTP1B by immunoblotting with an antiserum that specifically recognizes dimedonylated cysteinyl residues[20]. Dimedone labeled PTP1B best at low $H_2O_2$ concentrations, but did not react with PTP1B in the presence of pervanadate (Supplementary Fig. 1b). These data support the previous conclusion that dimedone reacts only with reversibly oxidized thiols[18]. Notably, pre-incubating purified PTP1B in 80 μM $H_2O_2$ (which favors reversible oxidation) before labeling diminished the efficiency of dimedonylation, compared with simultaneous oxidation and dimedone-labeling (Supplementary Fig. 1c). Previous reports have shown that, upon initial oxidation, the cysteine-sulfenic acid of PTP1B (PTP1B-SOH) is converted to an intramolecular sulfenylamide bond (PTP1B-SN) in PTP1B crystals[8] and in cells[21]. Taken together with these results, our biochemical data suggest that dimedone preferentially labels reversibly oxidized PTP1B in the PTP1B-SOH state, and labels the PTP1B-SN form poorly if at all (see Discussion).

We used anti-dimedone-Cys antiserum to immunostain oxidized proteins in Swiss 3T3 mouse fibroblasts. Fibroblasts were exposed to PDGF (50 ng ml$^{-1}$), $H_2O_2$ (1 mM) or left untreated, and then labeled with dimedone (5 mM) for 5 min while undergoing paraformaldehyde-fixation. Fixed, dimedone-labeled cells were then immunostained with anti-dimedone-Cys antiserum. Consistent with an earlier report[20], dimedone-labeled proteins were detected throughout cells, and total fluorescence intensity was increased slightly by $H_2O_2$ treatment (Supplementary Fig. 1d). By contrast, PDGF-stimulation did not cause a detectable change in the total intensity of anti-dimedone-Cys immunolabeling, most likely reflecting a low amount of oxidized proteins in response to growth factor, compared with $H_2O_2$ (Supplementary Fig. 1d). Treatment of cells with excess $H_2O_2$ (50 mM) diminished anti-dimedone-Cys antibody labeling, again supporting the selectivity of dimedone for reversibly oxidized proteins (Supplementary Fig. 1e). Of note, anti-dimedone-Cys immunoblots of lysates from dimedone-labeled Swiss 3T3 cells (Supplementary Fig. 1f) revealed that SHP2 is not a major dimedone-labeled protein in these cells. Likewise, there was no appreciable change in overall fluorescence signal when immortalized mouse embryo fibroblasts (MEFs) with or without SHP2[22] were labeled with dimedone and immunostained with anti-dimedone-Cys antibodies (Supplementary Fig. 1g).

Therefore, we attempted to "optically purify" the oxidized SHP2 (ox-SHP2) signal from the large population of oxidized (and subsequently, dimedonylated) proteins in cells. To do so, we combined dimedone-labeling/anti-dimedone-Cys antibody binding with anti-SHP2 antibody binding, followed by PLA[23] (hereafter, dimedone-PLA). PLA detects any two antibodies that co-localize within ~ 40 nm[23], in our method, anti-dimedone-Cys and anti-SHP2 antibodies, respectively (Fig. 1a). Dimedone-PLA for SHP2 visualized ox-SHP2 as puncta; notably, the number

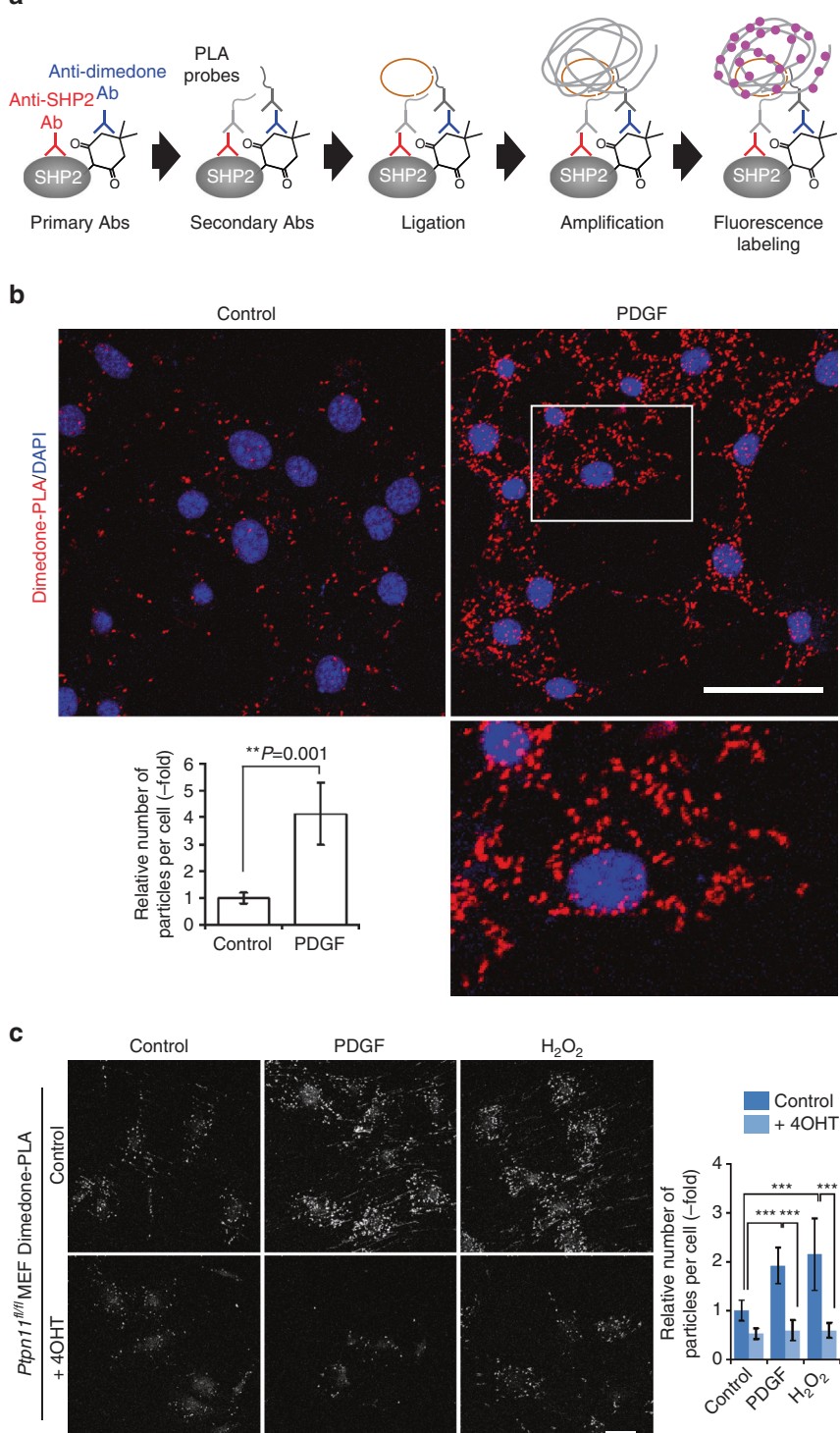

**Fig. 1** Detection of oxidized SHP2 by dimedone-PLA. **a** Schematic illustrating method for detecting oxidized SHP2 by dimedone-PLA. **b** Serum-starved Swiss 3T3 cells were stimulated with PDGF-BB (50 ng ml$^{-1}$) for 10 min or left unstimulated. Cells were fixed in the presence of dimedone (5 mM) for 5 min, and subjected to dimedone-PLA (*red*). Nuclei were stained with DAPI (*blue*). *Representative images* are shown for each condition from one of >4 independent biological replicates. A higher magnification image of the *boxed region* is shown at the *bottom right*. The *graph* shows the average number of PLA signals per cell (*n* = 6 images for each condition, 5–20 cells in an image), relative to control cells without stimulation (set as 1). The *P*-value was calculated using a two-tailed Welch's *t*-test. *Error bars* represent SD. **c** Serum-starved *Ptpn11*$^{fl/fl}$ MEFs expressing CRE-ER$^{Tam}$ treated with or without 4-hydroxytamoxifen (*4OHT*) were stimulated with PDGF-BB (50 ng ml$^{-1}$) or H$_2$O$_2$ (1 mM) for 10 min. Cells were fixed in the presence of dimedone, and subjected to dimedone-PLA (*gray*). *Representative images* are shown for each condition from one of three independent experiments. The *graph* shows average number of PLA signals per cell (*n* = 6 images for each condition, 5–20 cells in an image), relative to control cells without stimulation (set as 1). ***P* < 0.0001, ANOVA with Bonferroni/Dunn's post-hoc test. *Error bars* represent SD. *Scale bars*: 50 μm

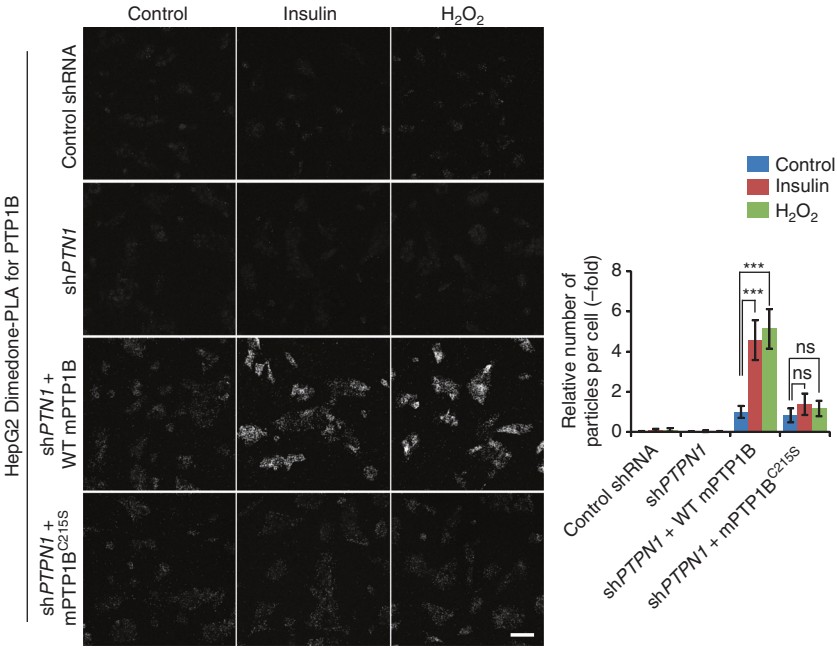

**Fig. 2** Detection of oxidized PTP1B by dimedone-PLA. Serum-starved HepG2 cells expressing control shRNA or shRNA targeting human *PTPN1* (sh*PTPN1*), with or without stable expression of either mouse WT PTP1B or PTP1B$^{C215S}$, were stimulated with insulin (25 nM) or $H_2O_2$ (1 mM) for 5 min or left unstimulated. Cells were then subjected to dimedone-PLA with anti-dimedone-Cys and anti-mouse PTP1B antibodies (*gray*). *Representative images* are shown for each condition from one of two independent biological replicates. The *graph* shows the average number of PLA signals per cell ($n = 6$ images for each condition, 80–120 cells in an image), relative to control cells without stimulation (set as 1). ***$P < 0.0001$, *ns* not significant, ANOVA with Bonferroni/Dunn's post-hoc test. *Error bars* represent SD. *Scale bar*: 50 μm

of puncta increased markedly in cells stimulated with PDGF-BB (50 ng ml$^{-1}$), EGF (50 ng ml$^{-1}$), or $H_2O_2$ (1 mM) for 10 min (Fig. 1b and Supplementary Fig. 2a). Dimedone-PLA puncta were observed mainly in the cytoplasm (Fig. 1b), even though SHP2 is found both in the cytoplasm and the nucleus[24].

Several lines of evidence indicate that these puncta represent ox-SHP2. First, the signals were eliminated in cells that did not undergo dimedone-labeling or that were incubated without anti-dimedone-Cys or anti-SHP2 antibodies (Supplementary Fig. 2b). Likewise, almost no puncta were observed in *Ptpn11*$^{-/-}$ MEFs, generated by Cre recombinase-mediated excision of a conditional (floxed) *Ptpn11* allele[22] (Fig. 1c). Re-expression of wild type (WT) SHP2, but not SHP2 bearing a C459E mutation (SHP2$^{C459E}$) that alters the cysteinyl residue in the SHP2 "signature motif," restored ROS-dependent puncta to *Ptpn11*$^{-/-}$ MEFs (Supplementary Fig. 2c, d). Depleting cellular ROS with *N*-acetyl-*L*-cysteine (NAC) suppressed dimedone-PLA signals. Conversely, lowering glutathione levels by treatment with the glutathione synthesis inhibitor *L*-buthionine-*S*,*R*-sulfoximine (BSO), enhanced the PLA signal (Supplementary Fig. 2e).

**Dimedone-PLA also detects PTP1B oxidation**. In principle, our method should be applicable to any reversibly oxidized protein, as long as its oxidation proceeds through a sulfenic acid state and there is specific antibody for the protein that can be used for PLA. Indeed, we detected oxidized PTP1B (ox-PTP1B) in $H_2O_2$-treated Swiss 3T3 fibroblasts by dimedone-PLA (Supplementary Fig. 3a). Notably, PDGF did not increase the number of puncta for ox-PTP1B. Previous studies have shown that PTP1B oxidation is evoked by insulin stimulation[21, 25]. As Swiss 3T3 fibroblasts respond weakly to insulin, we tested HepG2 cells (Supplementary Fig. 3b). HepG2 cells expressing short-hairpin RNA (shRNA) against endogenous *PTPN1*, which encodes PTP1B, were infected with a lentivirus harboring either WT mouse PTP1B (mPTP1B) or a serine substitution mutant of the catalytic cysteine residue

(mPTP1B$^{C215S}$) (Supplementary Fig. 3c, d). As expected, insulin stimulation evoked a significant increase in dimedone-PLA signal in WT mPTP1B-expressing HepG2 cells, but not in cells expressing mPTP1B$^{C215S}$ (Fig. 2). These results indicate that our method is not restricted to ox-SHP2 detection, but is applicable to other PTPs, and likely non-PTPs as well.

**Localization of PDGF-evoked SHP2 oxidation**. Upon stimulation with appropriate growth factors, RTKs, including the PDGF receptor (PDGFR), are activated, trans-phosphorylated, and recruit multiple signal relay proteins containing SH2 or phosphotyrosine-binding domains, including SHP2[26, 27]. Activated RTKs are endocytosed into clathrin-coated vesicles, clathrin dissociates, and transport occurs sequentially into early endosomes, late endosomes, and finally to lysosomes to be degraded. Some endocytosed RTKs can be recycled back to the plasma membrane[27, 28].

We asked when and where SHP2 oxidation occurs during this sequence. Stimulation of Swiss 3T3 cells with PDGF or EGF (50 ng ml$^{-1}$ each) led to a transient increase of ox-SHP2 to 2–3-fold over basal levels. Ox-SHP2 levels peaked at 5–10 min, and then decreased to ~ 1.5-fold basal levels by 60–120 min (Fig. 3a and Supplementary Fig. 4a). These findings are consistent with previous biochemical studies that reported SHP2 oxidation in response to PDGF stimulation[15], although EGF-induced SHP2 oxidation has not been reported heretofore. Surface PDGFRβ was endocytosed rapidly, with decreased surface immunofluorescence detectable within 2.5 min after PDGF stimulation (Supplementary Fig. 4b). Hence, maximal SHP2 oxidation occurs at an early stage of PDGFR trafficking.

We next analyzed the dynamics of PDGF-induced SHP2 oxidation in space and time by co-staining dimedone-PLA-labeled cells with antibodies against PDGFRβ, clathrin heavy chain (CHC), and the early endosome markers RAB5 and EEA1 at various times after PDGF addition. PDGFR, CHC, and

RAB5 fluorescence also localized to puncta, which overlapped with ox-SHP2 signals at different times after stimulation (Fig. 3b *left*). PLA generates relatively sparse, large and pleomorphic puncta, reflecting the extended DNA strands that amplify the fluorescence signals. Typical co-localization metrics, such as Pearson's or Manders' coefficients, are not appropriate for assessing co-localization between PLA signals and immunostained intracellular structures, as these coefficients assume that the intensity in each pixel represents a large number of molecules

(also see Methods). To determine whether the ox-SHP2 signals were randomly (i.e., stochastically) distributed within cells, or rather were in association/proximity with specific intracellular compartments, we developed an object-based image analysis. The punctate signals for ox-SHP2 and each marker were segmented, the center of mass for each puncta was determined, and the distance of each ox-SHP2 signal to the nearest marker signal was quantified. Notably, the median distance from the ox-SHP2 puncta to the PDGFRβ or CHC signals showed a

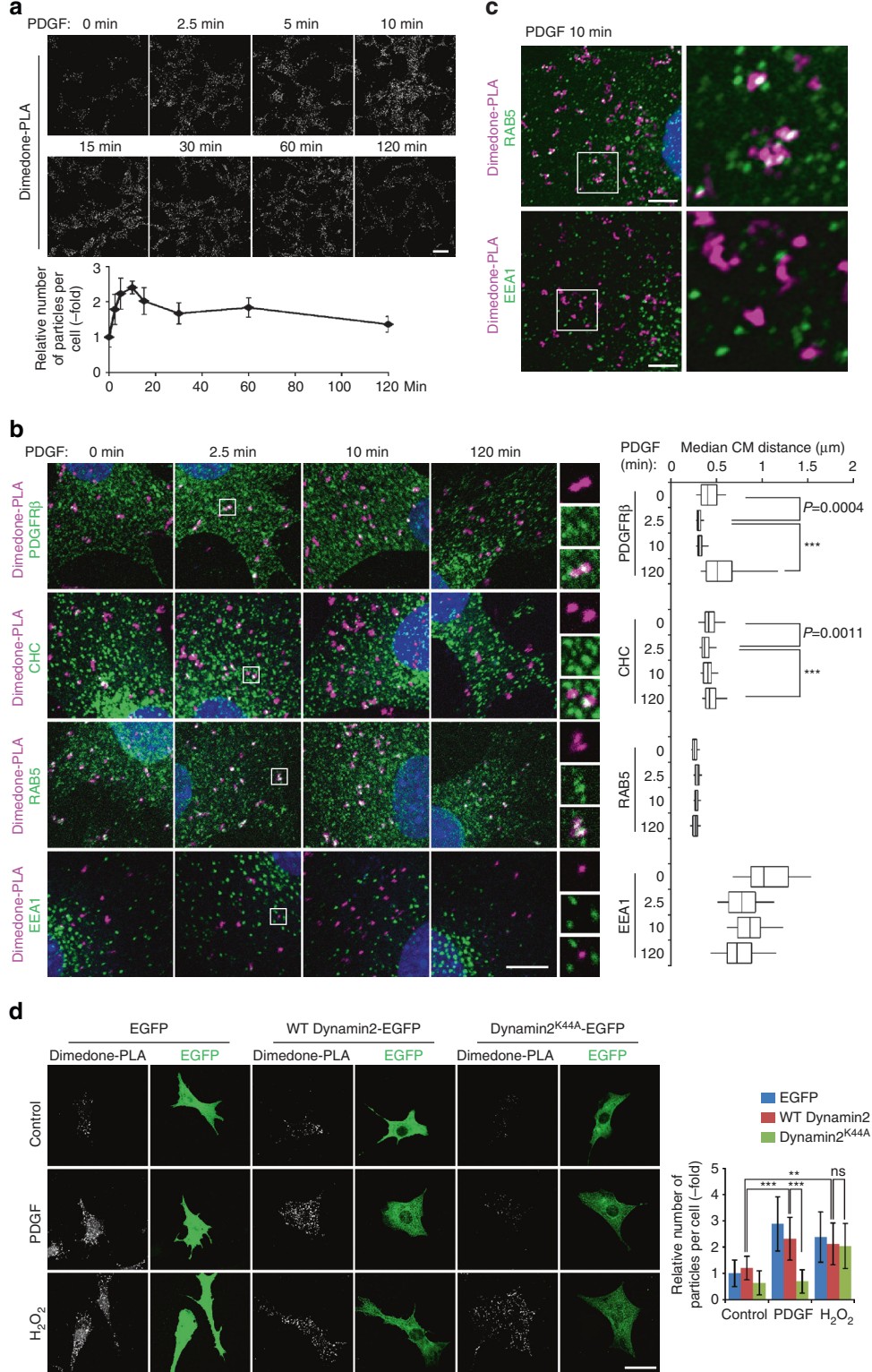

transient decrease at 2.5 min after PDGF stimulation to ~ 0.30 or ~ 0.40 μm, respectively, whereas the distance from ox-SHP2 to RAB5 remained at ~ 0.30 μm over the entire 120 min observation period (Fig. 3b *right*). The median radius of the ox-SHP2 puncta themselves was ~ 0.55 μm, which is larger than the distance between ox-SHP2 and the PDGFRβ, CHC, and RAB5, respectively. Furthermore, the distance between PDGFRβ, CHC, and RAB5 signals, respectively, was larger than the distance between ox-SHP2 and each of these molecules (Supplementary Fig. 4c). By contrast, the median distance between ox-SHP2 and the nearest EEA1 signal was ~ 0.85 μm over the entire observation period, which was comparable to the median distance between each EEA1 punctum (Fig. 3b and Supplementary Fig. 4c). These data indicate that ox-SHP2 is transiently in close proximity to PDGFRβ and CHC at early times after PDGF stimulation, RAB5 and ox-SHP2 are constitutively in proximity, and ox-SHP2 does not co-localize with EEA1. Semi-super resolution microscopy (AiryScan) confirmed the adjacency of ox-SHP2 puncta with RAB5+, but not with EEA1+, vesicles (Fig. 3c). Consistent with these observations, ox-SHP2 did not co-localize with the late endosomal markers RAB7 and RAB9 or with the recycling endosomal marker RAB11[29] (Supplementary Fig. 4d). PLA signals generated by anti-extracellular signal-regulated kinase 2 (ERK2) and anti-phospho ERK antibodies were distributed throughout the cytoplasm and nucleus, but showed no specific co-localization with RAB5 (Supplementary Fig. 4e). These results argue against artifactual co-localization of RAB5 with any punctate signal and for the specificity of ox-SHP2/RAB5 co-localization.

RAB5+ endocytic vesicles recruit EEA1 for tethering and fusion to early endosomes[29, 30]. Therefore, our results suggest that SHP2 binds to, and is endocytosed with, PDGFRβ (or rapidly associates with the receptor at the clathrin-coated vesicle stage), and undergoes oxidation during the clathrin coated-vesicle/early phase of the endocytic process before reaching EEA1+ early endosomes. Consistent with this model, immunostaining reveals the presence of a pool of RAB5+/EEA1-vesicles (Supplementary Fig. 4f). By contrast, ox-SHP2 signals do not colocalize with RAB5 in $H_2O_2$-treated cells (Supplementary Fig. 4g). Hence, $H_2O_2$-evoked SHP2 oxidation, unlike growth factor induced oxidation, does not proceed through an endosomal compartment and presumably can occur randomly in the cytoplasm. Failure of ox-SHP2 to co-localize with EEA1 at any time (Fig. 3b, c) presumably reflects reduction/re-activation of PDGFR-bound SHP2 at EEA1+ early endosomes or dissociation of ox-SHP2 from PDGFR.

To ask if growth factor-dependent receptor endocytosis is necessary for SHP2 oxidation, we assessed the effects of the dynamin inhibitor Dynasore[31] and the clathrin inhibitor Pitstop® 2[32]. Treatment of Swiss 3T3 cells with either inhibitor abolished the PDGF-dependent increase of ox-SHP2 without affecting $H_2O_2$-evoked ox-SHP2 generation (Supplementary Fig. 5a). Expression of dynamin2$^{K44A}$, which acts as a dominant negative mutant[33], also suppressed PDGF-dependent SHP2 oxidation (Fig. 3d and Supplementary Fig. 5b). Collectively, these data establish that clathrin/dynamin-dependent endocytosis is necessary for PDGF-dependent SHP2 oxidation. However, inhibiting endocytosis (either pharmacologically or with dominant negative dynamin2) also decreased overall PDGFRβ tyrosyl phosphorylation (Supplementary Fig. 5c, d). Therefore, it is unclear whether endocytosis is required to trigger SHP2 oxidation *per se* or merely is necessary for maximum PDGFR activation, and ultimately, PDGF-evoked ROS production.

**NOX complexes are necessary for PDGF-evoked SHP2 oxidation**. Since the first report of PDGF-induced ROS production by Finkel and co-workers[34], the source, location and pathway mediating ROS generation have remained unclear/controversial. One proposed mechanism involves growth factor-evoked activation of non-phagocytic NOX complexes, possibly via phosphoinositide 3-kinase (PI3K) and the small G protein RAC1[35, 36]. Other studies implicate mitochondria as the source of growth factor-evoked ROS, via a pathway involving p66SHC (Src homologous and collagen)[37]. NOX family members and the mitochondrial complexes I/III produce $O_2^-$, which is dismutated to $H_2O_2$ by superoxide dismutases[1]. Catalase specifically decomposes $H_2O_2$, and exogenous expression of a catalase mutant that localizes to the cytoplasm blocked PDGF-evoked SHP2 oxidation (Fig. 4a and Supplementary Fig. 6a). Hence, $H_2O_2$ is responsible for PDGF-dependent SHP2 oxidation. Furthermore, the NOX inhibitor diphenyleneiodonium (DPI) or the more selective inhibitor imipramine-blue[38] significantly suppressed PDGF-, but not $H_2O_2$-evoked SHP2 oxidation (Fig. 4b). By contrast, the mitochondria-targeted anti-oxidant MitoQ[39] had no apparent effect on PDGF-induced ox-SHP2 levels (Supplementary Fig. 6b). These results implicate NOX complex(es), rather than the mitochondrial electron transport chain, as the main source of ROS for PDGF-induced SHP2 oxidation. Consistent with a report implicating PI3K in PDGF-evoked ROS production[36], the PI3K inhibitors LY294002 or BKM120 antagonized PDGF-evoked ox-SHP2 generation (Supplementary Fig. 6c, d).

**Fig. 3** Spatio-temporal dynamics of SHP2 oxidation. **a** Serum-starved Swiss 3T3 cells were stimulated with PDGF-BB (50 ng ml$^{-1}$) for the indicated times, and subjected to dimedone-PLA. *Representative images* are shown for each condition from one of two independent experiments. The graph shows the average number of PLA signals per cell ($n = 6$ images for each condition, 5–20 cells in an image), relative to unstimulated control cells (normalized to 1). *Scale bar*: 50 μm. **b** Serum-starved Swiss 3T3 cells were stimulated with PDGF-BB (50 ng ml$^{-1}$) for the indicated times. Dimedone-PLA (*magenta*) and co-staining with the indicated markers (*green*) are shown. Nuclei were stained with DAPI (*blue*). *Representative images* are shown for each condition from one of three independent experiments. Higher magnification images of the *boxed regions* are shown. Median distances of centers of mass (*CM*) between punctate signals of ox-SHP2 and the nearest indicated marker signal in cells were obtained by object-based image analysis. The *box-whisker plots* show the median inter-object distances at the indicated times after stimulation ($n = 50$ cells, each time point). *Boxes* indicate the 25th–75th percentile; *whiskers* represent the 5th–95th percentile. ***$P < 0.0001$, ANOVA with Bonferroni/Dunn's post-hoc test. *Scale bar*: 10 μm. **c** Serum-starved Swiss 3T3 cells were stimulated with PDGF-BB (50 ng ml$^{-1}$) for 10 min, and subjected to dimedone-PLA (*magenta*) and co-staining with the indicated antibodies (*green*). Representative semi-super resolution microscopic images (AiryScan) and higher magnification images of the *boxed region* from one of two independent experiments are shown. *Scale bars*: 5 μm. **d** Swiss 3T3 cells expressing EGFP, EGFP-fused wild type dynamin2 (WT Dynamin2) or dominant-negative dynamin2 (dynamin2$^{K44A}$) were serum-starved and stimulated with PDGF-BB (50 ng ml$^{-1}$) or $H_2O_2$ (1 mM) for 10 min, and subjected to dimedone-PLA. *Representative images* of co-staining of dimedone-PLA (*gray*) and EGFP (*green*) are shown for each condition from one of two independent experiments. The *graph* represents the average number of PLA signals per cell ($n = 15$ cells), relative to unstimulated control cells (normalized to 1). ***$P < 0.0001$, *ns* not significant, ANOVA with Bonferroni/Dunn's post-hoc test. *Scale bar*: 50 μm. *Error bars* represent SD

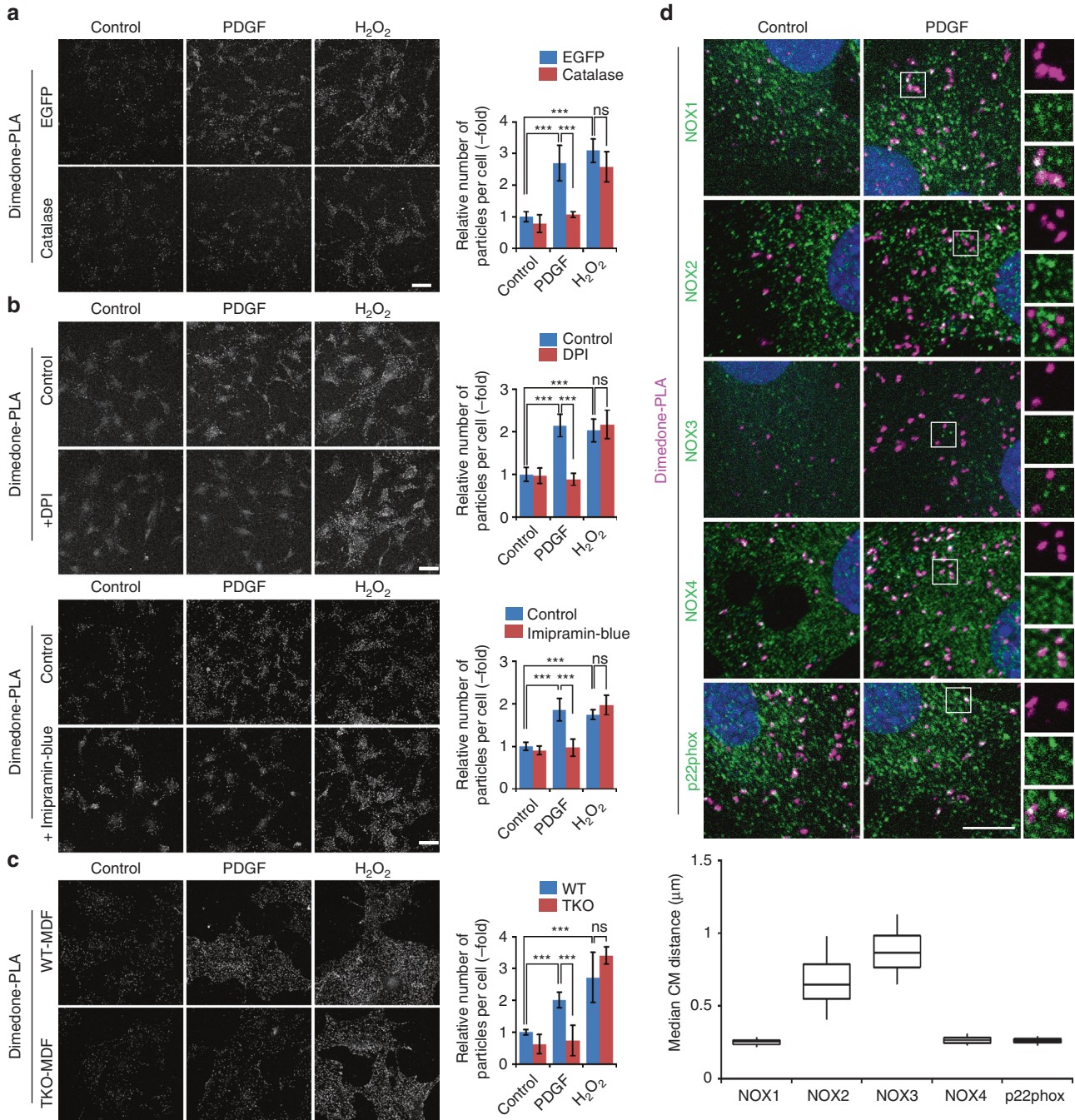

**Fig. 4** NOX complexes catalyze PDGF-evoked SHP2 oxidation. **a** Serum-starved Swiss 3T3 cells expressing EGFP or cytoplasmic catalase were stimulated with PDGF-BB (50 ng ml$^{-1}$) or H$_2$O$_2$ (1 mM) for 10 min and subjected to dimedone-PLA (*gray*). *Representative images* are shown for each condition from one of two independent experiments. **b** Serum-starved Swiss 3T3 cells were left untreated or were pre-treated with DPI (2.5 μM) (*top*) or imipramine-blue (6 μM) (*bottom*) for 5 min, stimulated with PDGF-BB (50 ng ml$^{-1}$) or H$_2$O$_2$ (1 mM) for 10 min and subjected to dimedone-PLA (*gray*). *Representative images* are shown for each condition from one of two independent experiments. **c** Serum-starved primary murine dermal fibroblasts (*MDFs*) from wild type (WT) or *Nox1, 2, 4* triple-KO (TKO) mice were stimulated with PDGF-BB (50 ng ml$^{-1}$) or H$_2$O$_2$ (1 mM) for 10 min and subjected to dimedone-PLA (*gray*). *Representative images* are shown for each condition from one of three independent experiments. **a–c** The *graphs* show the average number of PLA signals per cell (*n* = 6 images for each condition, 5–15 cells in an image), relative to control cells (normalized to 1). *Error bars* represent SD. ***$P$ < 0.0001, *ns* not significant, ANOVA with Bonferroni/Dunn's post-hoc test. *Scale bar*: 50 μm. **d** Serum-starved Swiss 3T3 cells were stimulated with PDGF-BB (50 ng ml$^{-1}$) for 10 min, and fixed in the presence of dimedone. Representative co-staining of ox-SHP2, detected by dimedone-PLA (*magenta*), and the indicated NOX components (*green*) are shown from one of three independent experiments (*top*). Nuclei were stained with DAPI (*blue*). Higher magnification images of the *boxed regions* are shown. *Scale bar*: 10 μm. Median distances of centers of mass (*CM*) between punctate signals of ox-SHP2 and the nearest indicated NOX protein signal in cells were obtained by object-based image analysis. *Box-whisker plots* show median distances at 10 min after PDGF stimulation (*n* = 50 cells). *Boxes* represent the 25th–75th percentiles and *whiskers* the 5th–95th percentiles

In mice, the NOX family comprises six transmembrane heme-containing proteins (NOX1-4 and DUOX1,2)[40]. Each NOX or DUOX forms a complex with other subunits (e.g., p22phox or DUOXA1/2) that regulate NOX activity. The topology of NOX/DUOX complexes predicts that if activation were to occur at the plasma membrane, then reduction of

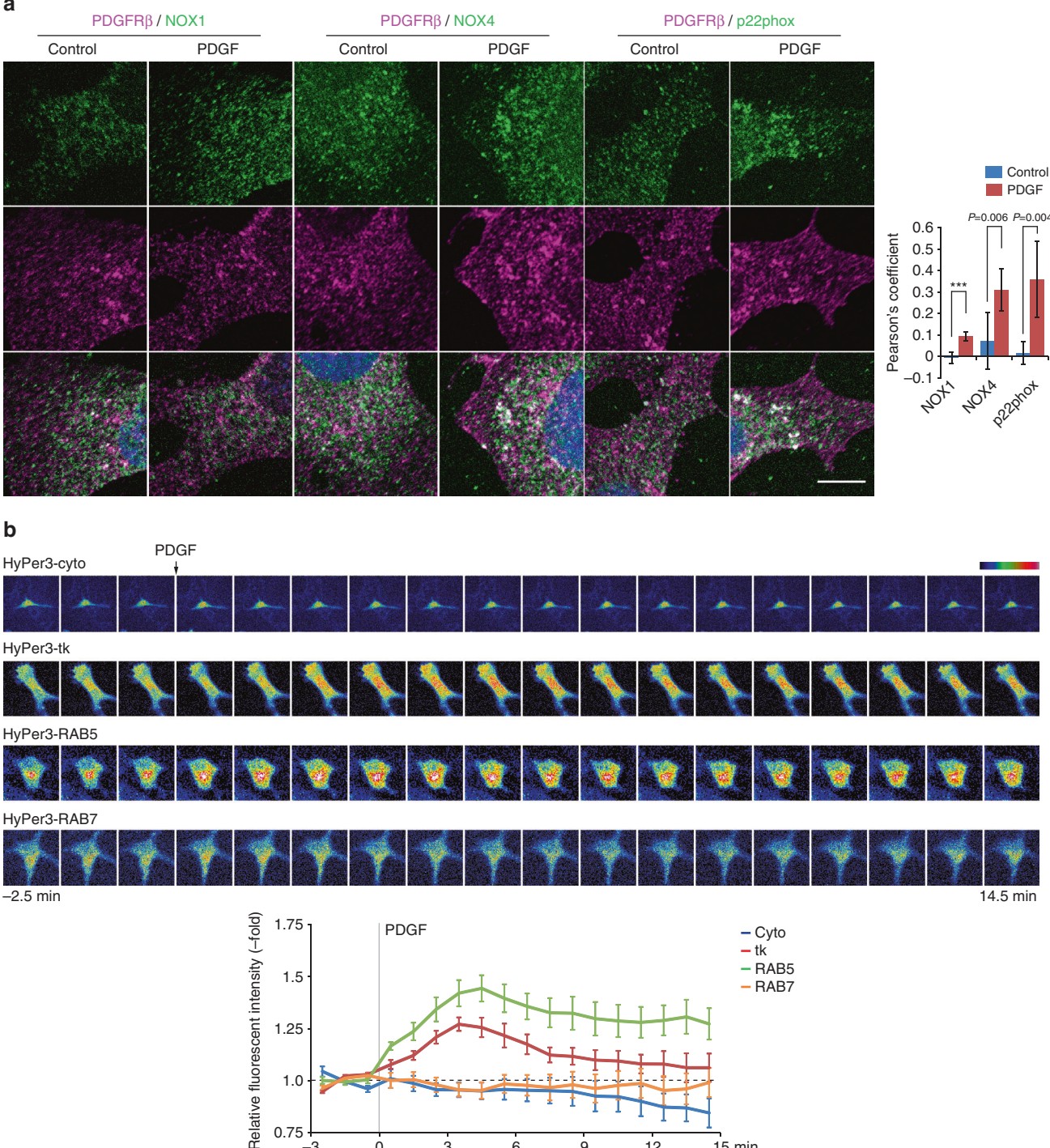

**Fig. 5** PDGF evokes redoxosome formation. **a** Serum-starved Swiss 3T3 cells were stimulated with PDGF-BB (50 ng ml$^{-1}$) for 5 min, then immunostained with anti-PDGFRβ (*magenta*) and the indicated anti-NOX component antibodies (*green*). *Representative images* are shown for each condition from one of two independent experiments. Nuclei were stained with DAPI (*blue*). *Scale bar*: 10 μm. The *graph* shows the average value of Pearson's coefficient for PDGFRβ and the indicated NOX components with or without PDGF-BB stimulation ($n = 1$ ROI from each of 10 cells) ***$P < 0.0001$; two-tailed Welch's *t*-test. **b** Serum-starved Swiss 3T3 cells expressing cytoplasmic HyPer3 (HyPer3-cyto), plasma membrane-targeted HyPer3 (HyPer3-tk), RAB5-fused HyPer3 (HyPer3-RAB5) or RAB7-fused HyPer3 (HyPer3-RAB7) were subjected to live-cell, time-lapse imaging. *Top panels*: Fluorescence intensity (pseudo-colored) of each HyPer3 construct shown at 1 min intervals. PDGF-BB (50 ng ml$^{-1}$) was added (*arrow*) after the third scan. *Bottom panel*: *Graph* shows the means of the relative fluorescence intensities of individual cells ($n = 7$ cells from each independent time-lapse experiment), setting the average of the first three time-points to 1 (*bottom*). *Error bars* represent SEM

NADPH to NADP$^+$ on the cytoplasmic side of membranes would be coupled to extracellular $O_2^-$ production. $O_2^-$ could move to the cytosol through a specific transporter or dismutate to $H_2O_2$, and then access the cytosol via channels or by diffusion[40], although such a process would seem to be inefficient.

To evaluate the involvement of specific NOXs in PDGF-evoked SHP2 oxidation, we used primary mouse dermal fibroblasts (MDFs) from wild type (WT-MDF) or *Nox1*, *Nox2*, and *Nox4*

triple-knockout animals (TKO-MDF) (ref. [41] and Supplementary Fig. 7a, b). Consistent with our inhibitor experiments, PDGF-, but not $H_2O_2$-evoked SHP2 oxidation was abrogated in TKO-MDFs, compared with their WT counterparts (Fig. 4c). Hence, PDGF-induced SHP2 oxidation requires NOX1, NOX2, and/or NOX4.

**Ox-SHP2 localizes with endosomes containing NOX proteins.** To obtain insight into which NOX complex(es) is/are involved in PDGF-dependent SHP2 oxidation, we co-stained ox-SHP2 with antibodies for each of the NOXs or p22phox. NOX proteins and p22phox showed punctate staining, some of which overlapped with ox-SHP2 PLA puncta (Fig. 4d, *top*). Using object-based image analysis, we found that PDGF-evoked ox-SHP2 signals were closely associated with NOX1 (median distance ~ 0.26 μm), NOX4 (median distance ~ 0.27 μm) and p22phox (median distance ~ 0.26 μm), but not with NOX2 or NOX3 (median distances ~ 0.65 and ~ 0.87 μm, respectively) (Fig. 4d, *bottom*). Again, the median distances of ox-SHP2 puncta to NOX1, NOX4, or p22phox puncta were significantly less than those between NOX puncta themselves (Supplementary Fig. 7c). Proximity of ox-SHP2 to NOX1, NOX4, and p22phox was observed before and after PDGF stimulation (Supplementary Fig. 7c), and semi-super resolution microscopy confirmed the adjacency of ox-SHP2 to NOX1 and NOX4 (Supplementary Fig. 7d). NOX1 and NOX4 were not visualized at the plasma membrane, mitochondria or in EEA1+ vesicles, although we cannot exclude the possibility that a small fraction of these proteins is found in these structures. Nevertheless, immunostaining revealed NOX1/4 localization to vesicular-like structures that partially co-localized with the ER marker calnexin (Supplementary Fig. 7e). The specificity of each of these antibodies for their respective NOX protein was confirmed by comparing staining of WT- and TKO-MDFs (Supplementary Fig. 7b).

Taken together, our data implicate NOX1 and/or NOX4 in PDGF-evoked SHP2 oxidation. Because our experiments also show that SHP2 oxidation occurs on, or close to, RAB5+ endosomes (see Fig. 3b, c), the overlap of ox-SHP2, NOX1,

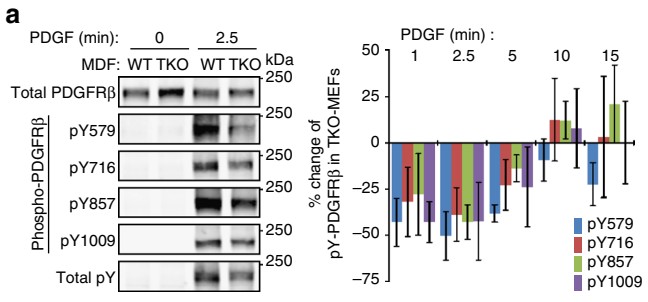

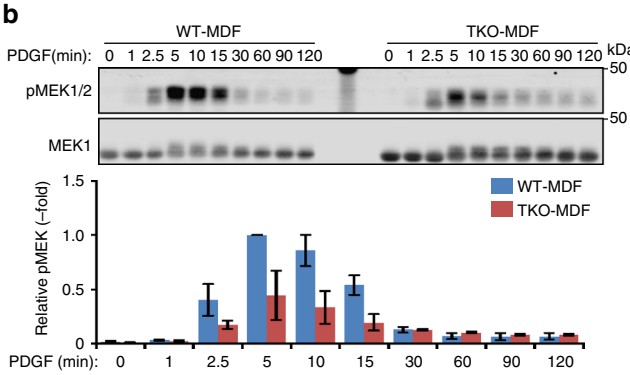

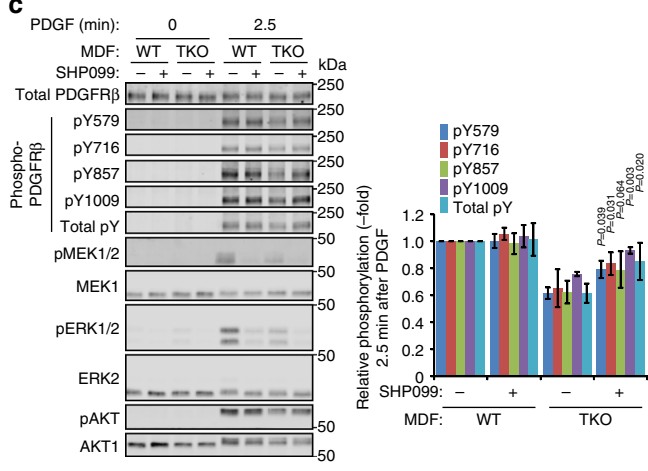

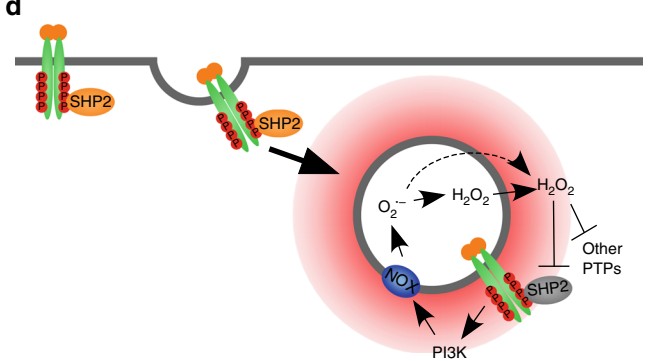

**Fig. 6** NOX activity is necessary for PDGF signaling. **a** Serum-starved WT- or TKO-MDFs were stimulated with PDGF-BB (50 ng ml$^{-1}$) for the indicated times. Lysates were subjected to immunoblotting with the indicated antibodies. *Representative immunoblots* are shown from one of four experiments. See also Supplementary Fig. 5a. *Graph at right* shows percentage change in phosphorylation of the indicated PDGFRβ tyrosine residues in WT- and TKO-MDFs at the indicated times post-stimulation, quantified from immunoblots ($n = 4$). *Error bars* represent SD. **b** Serum-starved WT- or TKO-MDFs were stimulated with PDGF-BB (50 ng ml$^{-1}$) for the indicated times. Lysates were subjected to immunoblotting with anti-pMEK1/2 and anti-MEK1 antibodies. *Representative immunoblots* are shown from one of four experiments. The *graph* shows the relative phosphorylation of MEK1, quantified from the immunoblots ($n = 4$), with the value in WT-MDFs at 5 min after PDGF stimulation set to 1. Data represent means ± SD. **c** Serum-starved WT- or TKO-MDFs were pre-treated with SHP099 (5 μM) or vehicle for 30 min, and stimulated with PDGF-BB (50 ng ml$^{-1}$) for 2.5 min. Lysates were subjected to immunoblotting with the indicated antibodies. *Representative immunoblots* are shown from one of four experiments. The *graph* shows the relative phosphorylation of the indicated PDGFRβ tyrosine residues, quantified from immunoblots ($n = 4$), with the values from untreated WT-MDFs set to 1. Data represent means ± SD. Values for each phosphotyrosine residue in TKO-MDFs with or without SHP099 treatment were compared by paired two-tailed *t*-tests. **d** Schematic of PDGF-evoked redoxosome formation and SHP2 oxidation. See text for details

and/or NOX4 signals presumably occurs on, or close to, such vesicles, which also contain ligand-bound, endocytosed PDGFR. EGF-evoked SHP2 oxidation also occurred in close proximity to RAB5 and NOX1/4 (Supplementary Fig. 7f). These findings argue that SHP2 oxidation occurs on, or close to, endosomal structures marked by RAB5, but not EEA1, that promote localized ROS generation, consistent with "redoxosomes"[2]. Although our object-based analysis has insufficient resolution to determine whether SHP2 oxidation actually occurs on (vs. close to) these structures, we note that earlier biochemical studies indicated that SHP2 oxidation was strictly associated with the PDGFR (which, in turn, would be on endocytosed vesicles)[15].

**PDGF-evoked redoxosome formation.** The "redoxosome" model proposes that IL-1 or TNF-α generates NOX-containing, redox-active endosomes wherein ROS generation is directed into membrane-enclosed vesicles (as a consequence of NOX topology)[2]. Our data suggest that PDGF stimulation also promotes redoxosome generation; i.e., NOX1 and/or NOX4 are recruited following PDGFR endocytosis, contribute to ROS production at RAB5+ endosomes, and result in oxidation of SHP2 (and possibly other targets). Indeed, NOX1 and NOX4 showed partial, but significant co-localization with RAB5 (Supplementary Fig. 8a). When calculated by Manders' coefficient, RAB5 co-localization with NOX1/NOX4 decreased slightly upon PDGF stimulation, although substantial co-localization remained (Supplementary Fig. 8a). On the other hand, NOX4 and p22phox, and to a lesser extent, NOX1, showed strongly increased co-localization with PDGFRβ at 5 min after PDGF stimulation (Fig. 5a), indicating recruitment of NOX proteins to PDGFR-containing vesicles.

To localize PDGF-evoked ROS generation, we utilized the sensor HyPer3[42], targeted selectively to different intracellular locales. HyPer3 is a circularly permuted yellow fluorescent protein (cpYFP) integrated into the regulatory domain of the $H_2O_2$-sensing protein OxyR, which exhibits increased fluorescence intensity in response to increased local concentrations of $H_2O_2$[42]. Time-lapse imaging of Swiss 3T3 fibroblasts showed that the fluorescence intensity of HyPer3 fused to the KRAS C-terminal sequence (HyPer3-tk), which is targeted to the plasma membrane, as well as that of HyPer3 fused to RAB5 (HyPer3-RAB5) such that the sensor faces the cytosol on RAB5+ vesicles, increased within 3–6 min of PDGF stimulation (Fig. 5b). By contrast, there was no detectable increase in the fluorescence of cytoplasmic HyPer3 (HyPer3-cyto) or HyPer3 fused with RAB7 (HyPer3-RAB7) in response to PDGF (Fig. 5b). Notably, EGFP, which is not sensitive to $H_2O_2$, fused to the KRAS C-terminal sequence or to RAB5 showed no PDGF-evoked change in fluorescence (Supplementary Fig. 8b). Furthermore, PDGF did not evoke significant ROS production at RAB5+ endosomes in TKO-MDFs (Supplementary Fig. 8c).

Collectively, these data strongly support a model in which PDGF-stimulation evokes formation of redoxosomes, recruiting and/or activating NOX complexes adjacent to the endocytosed PDGFR. Recruited NOX complexes catalyze ROS generation at these RAB5+ endosomes, resulting in spatially limited SHP2 oxidation. Alternatively, SHP2-containing RAB5+/EEA1- vesicles could be recruited adjacent to, and possibly in contact with, NOX1/4-containing regions of the ER, resulting in exposure of RAB5+ vesicles/SHP2 to ER-localized ROS generation.

**Redoxosomes prevent precocious PDGFRβ dephosphorylation.** To assess the physiological role of PDGF-evoked redoxosome formation/SHP2 oxidation, we monitored the phosphorylation of PDGFRβ and its downstream signaling components in TKO-MDFs. PDGFRβ was degraded with similar kinetics following PDGF stimulation of WT- and TKO-MDFs (Supplementary Fig. 9a, b). In WT-MDFs, the relative tyrosyl phosphorylation of PDGFRβ peaked at around 10 min after stimulation, followed by a decay of phosphorylation at 30–120 min (Supplementary Fig. 9a, c). Compared with WT-MDFs, TKO-MDFs showed significantly decreased phosphorylation of Y579, Y716, Y857, and Y1009 in PDGFRβ at early times after ligand stimulation (1 and 2.5 min), but comparable phosphorylation levels at later times (10 min and later; Fig. 6a and Supplementary Fig. 9a, c). These phosphorylation differences correlate temporally with ROS production at RAB5+ vesicles as detected by HyPer3 (Fig. 5b), ox-SHP2 generation (Fig. 3a), and ox-SHP2 localization near the PDGFRβ (Fig. 3b). Tyr-857 is an auto-phosphorylation site, which is necessary to enhance the kinase activity of PDGFRβ, whereas Y579, Y716, and Y1009 are reported binding sites for SHC/SRC/ signal transducer and activator of transcription 5 (STAT5), growth factor receptor-binding protein 2 (GRB2)/SOS and phospholipase Cγ (PLCγ)/SHP2, respectively[27]. Consistent with decreased phosphorylation of the latter PDGFRβ tyrosyl residues, phosphorylation of MEK1 (MAP kinase or ERK kinase-1) at S217 and S221 was decreased significantly in TKO-MDFs (Fig. 6b). Importantly, treatment with the allosteric SHP2 inhibitor SHP099[43] partially restored PDGFRβ tyrosine phosphorylation in TKO-MDFs, while having almost no effect on WT-MDFs (Fig. 6c). These data suggest that redoxosome generation spatially and temporally inhibits SHP2 and other PTPs to enable the proper level of PDGFR activation specifically at early times after ligand stimulation.

Notably, PDGF-dependent activation of ERK was suppressed by SHP099 in WT- and TKO-MDFs, regardless of PDGFRβ phosphorylation/activation (Fig. 6c). Hence, SHP2 has spatially distinct actions on negative regulation of the PDGFR, which appears only in the absence of NOX1/2/4, and on positive regulation in RAS/ERK pathway activation, respectively.

## Discussion

Biochemical evidence has argued for an important role for growth factor-induced ROS production in cellular signaling[1] and for PTPs as critical ROS targets[6, 7]. Yet with the exception of a conformation-specific scFv (single-chain variable fragment) that selectively detects the sulfenylamide form of PTP1B[21], there has been no way to visualize oxidized PTPs or other proteins within cells. We capitalized on the preferential reactivity of dimedone for the SOH form of oxidized protein-thiols[17, 18], the availability of anti-dimedone-Cys antiserum[20] and the highly sensitive and specific PLA method[23] to visualize growth factor-evoked SHP2 oxidation.

Our experiments confirm that dimedone prefentially labels reversibly oxidized PTPs, most likely in the PTP-SOH state (Supplementary Fig. 1a–c). PTP-sulfenic acids convert to intramolecular sulfenylamides[8] or disulfides[44] in vitro, and the sulfenylamide state has been detected in cells by a PTP1B-SN-specific scFv[21]. However, the kinetics and stoichiometry of these conversions are unclear, due to our inability to monitor each of these forms in cells. Accordingly, these observations do not exclude the possibility of a pool of the PTP-SOH form. Indeed, our in vitro dimedone-labeling experiments, combined with our dimedone-PLA results, strongly suggest that PTP-SOH species exist in cells.

Most importantly, our results show that SHP2 oxidation is regulated in space and time, and provide support for the "redoxosome" model[2]. We propose that PDGF-dependent redoxosome formation and SHP2 oxidation occur as follows: (1) Upon PDGF stimulation, PDGFR is activated, phosphorylated

and recruits SHP2 and other signaling molecules; (2) PDGFR is endocytosed in a clathrin/dynamin-dependent manner, and recruits NOX1/4 at the endosomes; (3) PDGFR and NOX-containing vesicles, via a PI3K activity-dependent process, produce $O_2^-$ in their interiors; (4) $O_2^-$ is either dismutated inside the vesicle or diffuses to the cytoplasm to be dismutated, in either case generating a localized $H_2O_2$ gradient; and (5) SHP2 bound to PDGFR and possibly other PTPs at/close to the redoxosome are oxidized by localized $H_2O_2$, resulting in temporal inactivation of PTP activity around PDGFR (Fig. 6d). Although PDGF evokes co-localization of PDGFRβ and NOX proteins (Fig. 5a), how NOX1/4 is recruited to PDGFR-containing endosomes remains to be elucidated. Conceivably, a small amount of plasma membrane-associated NOXs are co-endocytosed with PDGFR to form redoxosomes. Alternatively, a fraction of NOX proteins reside on cytoplasmic vesicular structures, and these vesicles are recruited and possibly fuse with PDGFR-containing endocytic vesicles. We also cannot exclude the possibility that endocytosing vesicles are close enough to the ER or other structures where NOXs reside and produce ROS.

The "redoxosomes" in IL-1 or TNF-α signaling are reportedly RAB5+/EEA1+[2, 45]. Redoxosome-produced ROS is required for TNF receptor-associated factor 6 (TRAF6) recruitment to the endocytosed IL-1 receptor complex and for TRAF2 recruitment to the TNF receptor complex, and consequently, for nuclear factor-κB activation in response to these ligands[2]. However, the critical target molecule(s) of ROS-evoked oxidation for TNF-α/IL-1 signal transduction has(ve) not been identified. Although PDGF-stimulation also induces structures consistent with redoxosomes, these structures are RAB5+/EEA1- (Fig. 3b, c). Whether RTK- and cytokine-evoked redoxosomes form in distinct locations, or instead, these discrepancies reflect technical differences, requires clarification. Notably, the earlier experiments could have enabled labeling of vesicles that contain RAB5 and EEA1 even if ROS production had occurred only in the RAB5+/EEA1- compartment[45]. Importantly, our assay enabled us to visualize a key signal relay molecule, SHP2, undergoing physiological oxidation on, or in close proximity to, these structures, providing direct evidence of spatially limited redoxosome-dependent protein oxidation. Furthermore, our results suggest that redoxosome-mediated oxidation events are likely to be common to a wide variety of receptor systems.

In the absence of ROS production at redoxosomes (i.e., in TKO-MDFs), failure to inhibit SHP2 and/or other PTPs at endocytic vesicles diminishes PDGFR and MEK activation in response to PDGF (Fig. 6). However, genetic and biochemical evidence implicate SHP2, and in particular, its catalytic activity, as a positive (i.e., signal enhancing) regulator of growth factor-dependent RAS/ERK activation[13, 14]. Previous work also indicates that SHP2 dephosphorylates the PDGFR[15]. Our observation that PDGFR-bound SHP2 is oxidized/inactivated at early time-points, together with the requirement of SHP2 activity for ERK activation, indicates that these two roles for SHP2 are spatially segregated and differentially regulated by ROS. Presumably, critical SHP2 substrates for RAS/ERK activation are accessed either before or after redoxosome action or at a distinct intracellular location. Recent data also suggest that SHP2 could have distinct functions proximal to, and distal from, EGFR activation[46].

Our results indicate NOX1/4-dependent SHP2 oxidation occurs in close proximity to specific endosomes in growth factor-stimulated fibroblasts. Previous reports showed that NOX2 is responsible for SHP2 oxidation in erythropoietin-stimulated endothelial progenitors[47] and that NOX4 is necessary for shear stress-evoked SHP2 oxidation in aortic endothelial cells[48], but those studies could not provide spatial information. These

findings do, however, imply that different NOX(s) catalyze SHP2 oxidation in different cells or signaling contexts, perhaps dependent on differential NOX expression. Another report argued that p66SHC and mitochondria are critical for PDGF-evoked ROS production and oxidation of PTPs, including SHP2[37]. Although strong suppression of oxidized PTPs was observed in p66SHC-depleted cells when compared with normal cells, p66SHC-depleted cells still showed a PDGF-dependent increase of PTP oxidation in their experiments. Hence, PDGF might stimulate p66SHC/mitochondria-dependent and -independent ROS production.

Multiple PTPs are purported targets of ROS-catalyzed oxidation[1, 6, 7] and PTPs have multiple, and distinct, functions in cell signaling[3]. Restricting their oxidation spatially and temporally is likely to be of critical importance, and visualizing such events is of clear interest. Yet with the exception of PTP1B oxidation in response to insulin stimulation[21], visualization of intracellular oxidation events has not been reported heretofore. Dimedone-PLA can also recognize insulin-evoked ox-PTP1B (Fig. 2), but because it capitalizes on the intrinsic chemistry of cysteine oxidation, our method is, in principle, adaptable to all PTP superfamily members, as well as any protein-thiols for which an appropriate antibody is available.

## Methods

**Cell culture.** Swiss 3T3 (3T3 Swiss albino) cells, *Ptpn11*[fl/fl] MEFs expressing CRE-ER[Tam22], and HepG2 cells were cultured in Dulbecco-modified Eagle's medium (DMEM), supplemented with 10% fetal bovine serum (FBS). To induce deletion of *Ptpn11*, *Ptpn11*[fl/fl] MEFs were treated with 1 μM 4-hydroxytamoxifen (4OHT) for 4 days. Primary MDFs[41] were cultured in DMEM/F-12 supplemented with 10% FBS. Swiss 3T3 cells and HepG2 cells were from a Neel lab-maintained stock, and were authenticated by the CellCheck short tandem repeat profiling service (IDEXX BioResearch). All cells were tested for mycoplasma using the Mycoplasma Plus detection kit (Agilent).

**Antibodies.** Mouse monoclonal anti-SHP2 (B-1, sc-7384), anti-PDGFRβ–pY716 (F-10, sc-365464), anti-ERK2 (D-2, sc-1647), and anti-RAB5A antibody (E-11, sc-166600), as well as rabbit polyclonal anti-SHP2 (C-18, sc-280), anti-PDGFRβ (958, sc-432), anti-PDGFRβpY579 (sc-135671), and anti-calnexin (H-70, sc-11397), goat polyclonal anti-PDGFRβ (M-20, sc-1627), anti-clathrin heavy chain (C-20, sc-6579), anti-RAB5A (P-12, sc-26566), anti-EEA1 (C-15, sc-6414), anti-NOX1 (H-15, sc-5821), anti-NOX2 (C-15, sc-5827), anti-NOX3 (A-12, sc-34699), anti-NOX4 (N-15, sc-21860), and anti-p22phox (C-17, sc-11712) antibodies were purchased from Santa Cruz Biotechnology. Mouse monoclonal anti-MEK1 (61B12, #2352) and anti-AKT1 (2H10, #2967), rabbit monoclonal anti-PDGFRβpY857 (C43E9, #3170), anti-PDGFRβpY1009 (42F9, #3124) and anti-pAKT T308 (244F9, #4056), as well as rabbit polyclonal anti-EEA1 (C45B10, #3288), anti-pMEK S217/221 (#9121), and anti-pERK1/2 T202/Y204 (#9101) antibodies were purchased from Cell Signaling. Mouse monoclonal anti-phosphotyrosine antibody cocktail (4G10 platinum, 05-1050) and rabbit anti-dimedonylated cysteine antiserum (anti-cysteine sulfenic acid, 07-2139) were purchased from Millipore. Mouse monoclonal anti-PTP1B (ab201974) and chicken polyclonal anti-GFP (ab13970) antibodies were purchased from Abcam. Goat polyclonal anti-PTP1B (AF3954) and anti-PDGFRβ (AF1042) antibodies were purchased from R&D Systems. Mouse anti-FLAG antibody (M2, F1804) was purchased from Sigma. Alexa fluor-conjugated secondary antibodies (Thermo Fisher) were used for immunostaining. All antibodies were used at the concentrations recommended by their manufacturers (1:1000 for immunoblotting and 1:500 for immunostaining), except for anti-SHP2, anti-PTP1B antibodies and anti-dimedonylated cysteine antiserum for dimedone-PLA (see below).

**Growth factors and chemical compounds.** Recombinant human PDGF-BB and EGF were purchased from Peprotech. 5,5-Dimethyl-1,3-cyclohexanedione (dimedone), *para*-nitrophenyl phosphate (*p*NPP), 4OHT, NAC, L-buthionine-S, R-sulfoximine (BSO), and DPI were purchased from Sigma. Catalase (*Aspergillus niger*) was purchased from Millipore. Dynasore[31] was kindly provided by Dr T. Kirchhausen (Harvard Med. Sch.) or purchased from Sigma. Pitstop® 2[32] was purchased from Abcam. Imipramine-blue[38] was synthesized and kindly provided by Drs N. Patel and H.W. Pauls (Campbell Family Institute for Breast Cancer Research, Princess Margaret Cancer Center, University Health Network, Toronto, Canada). MitoQ (Mitoquinone)[39] was purchased from BioTrend. SHP099[43] was purchased from Alputon Inc. Rhodamine phalloidin was purchased from Thermo Fisher Scientific.

**Expression constructs**. Retroviral expression vectors for wild type SHP2 (WT SHP2) and C459E SHP2 (SHP2$^{C459E}$) were generated by subcloning human *PTPN11* complementary DNA (cDNA) into pMSCV-IRES-EGFP (Clontech). pSUPER-puro-based control shRNA or sh*PTPN1* retroviral expression vectors were designed to target 5′-CCGCCCAGAGGAGCTATATTC-3′ or 5′-CCGCCC AAAGGAGTTACATTC-3′, respectively, as reported[49]. Expression vectors for WT mPTP1B or mPTP1B$^{C215S}$ were generated by subcloning the appropriate cDNAs into the lentiviral expression vector pFB-neo. Expression vectors for EGFP-fused RAB5A, RAB7A, RAB9A, and RAB11A and pEGFP-WT Dynamin2 and Dynamin2$^{K44A}$ were purchased from Addgene. A cDNA of mouse catalase that lacks its peroxisome-targeting sequence was purchased from Addgene, and subcloned into pMSCV-IRES-EGFP, adding a start codon followed by a FLAG-tag encoding sequence at the 5′ end. Expression vectors for HyPer3-tk, HyPer3-RAB5, and HyPer3-RAB7 were generated by fusing cDNAs for either the human KRAS C-terminal sequence, human *RABA5A* or *RAB7A* (Addgene) to the 3′ end of *HyPer3* cDNA, respectively. All constructs were confirmed by DNA sequencing.

**Transfection, infection, and cell sorting**. Swiss 3T3 cells were transfected with the indicated expression vectors using Lipofectamine3000 (Thermo Fisher), according to the manufacturer's protocol. Twenty-four hours post-transfection, GFP-positive cells were isolated by FACS using a MoFlo (Beckman Coulter), and then were seeded for dimedone-PLA, immunostaining or immunoblotting. *Ptpn11*$^{fl/fl}$ MEFs expressing WT SHP2, or SHP2$^{C459E}$, and Swiss 3T3 cells expressing FLAG-catalase were generated by retroviral infection, according to the manufacturer's protocol (pMSCV retrovirus system, Clontech), followed by FACS for GFP-positive cells. HepG2 cells were infected stably with VSVG-pseudotyped retrovirus carrying control shRNA or sh*PTPN1*, and selected in puromycin. Cells were further infected with VSVG-pseudotyped retrovirus carrying expression vector for WT mPTP1B or mPTP1B$^{C215S}$, and selected in geneticin (G418).

**In vitro phosphatase assay and dimedone-labeling**. Purified recombinant human PTP1B (1–321) was kindly provided by Dr N.K. Tonks (Cold Spring Harbor Laboratory, New York). PTP1B (0.5 µg ml$^{-1}$) was pre-incubated in a buffer containing 12.5 mM Hepes (pH 7.4), 0.25 mM EDTA, and 35 µM DTT in the presence of 10 mM DTT, 80 µM $H_2O_2$, 320 µM $H_2O_2$ or 1 mM pervanadate for 30 min at room temperature (R.T.). To measure reversibility of oxidation, aliquots from these reactions were incubated further in the presence of 10 mM DTT and 5U of catalase at R.T. for 3 h. Catalytic activity was then measured by using *p*NPP as substrate and pre-incubated PTP1B (125 ng) in 200 µl of phosphatase assay buffer (25 mM Hepes pH 7.4, 50 mM NaCl, 10 mM *p*NPP) for 5 min at 37°C. For in vitro dimedone-labeling, purified PTP1B (0.25 µg µl$^{-1}$) was incubated for 5 min at R.T. in the presence or absence of 5 mM dimedone in labeling buffer (12.5 mM Hepes, pH 7.4, 0.25 mM EDTA and 35 µM DTT) containing a final concentration of 10 mM DTT, 80 µM $H_2O_2$, 320 µM $H_2O_2$ or 1 mM pervanadate, as indicated. Two microliters of each reaction were diluted into 148 µl of sodium dodecyl sulfate polyacrylamide gel electrophoresis (SDS-PAGE) sample buffer, and subjected to SDS-PAGE, followed by immunoblotting with anti-dimedone-Cys and anti-PTP1B antibodies. For post-oxidation dimedone-labeling, purified PTP1B (0.5 µg µl$^{-1}$) was first incubated in labeling buffer with 10 mM DTT, 80 µM $H_2O_2$, or 320 µM $H_2O_2$ at R.T. for 30 min, and then further incubated for 5 min at R.T. after addition of an equal amount of 10 mM dimedone solution containing the same concentration of DTT or $H_2O_2$.

**Dimedone-PLA**. Swiss 3T3 ($2 \times 10^4$) or HepG2 ($5 \times 10^4$) cells were seeded on 12 mm, poly-*L*-lysine-coated circular glass coverslips, and then were serum-starved for 16 h. After stimulation with the indicated growth factor or $H_2O_2$, cells were rinsed with PBS and fixed with 4% paraformaldehyde/PBS containing 5 mM dimedone/0.25% DMSO for 5 min at R.T. The dimedone-containing fixation solution was prepared fresh before each experiment and filtered through a PES membrane with pore size = 0.22 µm (Millipore). Cells were then rinsed six times with PBS, permeabilized with 0.1% TritonX-100/0.2% BSA/PBS for 10 min, and blocked in 1% BSA/PBS for 30 min. After incubation with primary antibodies (anti-SHP2 antibody: 1:5000 or anti-PTP1B: 1:2000, with anti-dimedonylated cysteine antiserum: 1:1000) for 1 h at R.T., cells were subjected to PLA (Duolink In Situ Detection Reagents Red, Sigma) with donkey anti-mouse (Duolink In Situ PLA Probe Anti-Mouse MINUS, Sigma) or donkey anti-goat (Duolink In Situ PLA Probe Anti-Goat MINUS, SIGMA) and anti-rabbit (Duolink In Situ PLA Probe Anti-Rabbit PLUS, SIGMA) secondary antibodies, according to the manufacturer's protocol. After the PLA procedure, cells were subjected to another round of sequential incubation with primary and Alexa fluorophore-conjugated secondary antibodies for co-staining with markers or NOX proteins. Coverslips were then mounted on glass slides using Prolongold containing DAPI (Thermo Fisher). Images were obtained by using an LSM700 confocal microscopy system (Carl Zeiss). Semi-super resolution microscopic images (AiryScan) were obtained by using an LSM880 (Carl Zeiss).

**Immunofluorescence**. Cells ($4 \times 10^4$ for anti-dimedone-Cys staining or $2 \times 10^4$ for other immunostaining) were seeded on coverslips as above, fixed in 4% paraformaldehyde/PBS for 10 min at R.T., permeabilized with 0.1% TritonX-100,

0.2% BSA/PBS for 10 min, and blocked with 1% BSA/PBS for 30 min, followed by sequential primary and secondary antibody treatments, as indicated. Images were obtained by using an LSM700 or an LSM880.

**Immunoblotting**. Cells were lysed in SDS lysis buffer (50 mM Tris-HCl pH7.5, 100 mM NaCl, 1 mM EDTA, 1% SDS, 2 mM $Na_3VO_4$), and subjected to SDS-PAGE, followed by transfer to Immobilon-FL PVDF membranes (Millipore). Membranes were incubated in 1% BSA/TBS containing 0.1% Tween20 for 30 min, and treated with primary antibodies in blocking buffer for 1 h, followed by treatment with IRDye-conjugated secondary antibodies (LI-COR). Images were obtained by using an ODYSSEY CLx quantitative IR fluorescent detection system (LI-COR), and quantified with Image Studio software Ver. 5.2 (LI-COR).

**Live-cell imaging**. Expression vectors for HyPer3 or its mutants were transfected into Swiss 3T3 fibroblasts using Lipofectamine 3000 (Thermo Fisher), as per the manufacturer's protocol. Twelve hours after transfection, cells were re-seeded ($1 \times 10^4$ cells/well) into glass-bottom 8-well lab-tech chambers (Nunc) or µ-Slide 8 Well (ibidi), and then serum-starved in phenol red-free DMEM for 16 h. The fluorescence intensities of HyPer3-expressing cells were monitored by using an LSM700 confocal microscopy system with a stage-top incubator at 37 °C in a 5% $CO_2$ atmosphere. Excitation was at 488 nm, and emission was monitored at 500–1000 nm. Images were obtained every 1 min, with PDGF added after the third scan. Cells that showed shrinking or detachment during the time-lapse were excluded from the analyses.

**Image analysis**. For dimedone-PLA images, the number of punctate signals generated by PLA and nuclei were counted from a single image by using the "analyze particle" function of FIJI/ImageJ software[50]. HepG2 cell nuclei were counted manually. The number of PLA signals per cell was calculated by dividing the particle number by the number of nuclei in an image, averaging six images per condition. To assess the localization of ox-SHP2 within intracellular compartments, we employed object-based analysis, which enabled us to compare one signal representing many molecules (e.g., RAB5, EEA1, etc.) with another (PLA) representing a resolution-limited, sparse number of molecules. For object-based image analyses, regions of interest (ROIs) were set according to cell shape using FIJI/ImageJ software[50], and the region of the nucleus (stained with DAPI) was masked using CellProfiler software[51]. Punctate signals from PLA (red) and marker staining (green) were segmented by using Otsu threshold algorithm in CellProfiler[51]. The centers of mass (CM) of the segmented objects were determined by using MATLAB software (The MathWorks, Inc.). To evaluate potential co-localization the nearest neighbor approach was applied by finding the distances between the CMs of each red/green object to the nearest of the green objects in each cell. The outliers in the obtained distances distributions were removed by using Tukey's method[52], and the median distances of red-to-green and green-to-green in each cell ($n = 50$ cells per condition) obtained using custom MATLAB scripts were subjected to statistical analyses. For pixel-based co-localization analyses, 225 µm$^2$ (Fig. 5a) or 56.25 µm$^2$ (Supplementary Fig. 8a) ROIs were set in cytoplasmic region of individual cells. Manders' and Pearson's coefficients were calculated by setting the same threshold for ROIs for samples stained with the same antibody, using the "coloc2" function in FIJI/ImageJ software[50]. For HyPer3 experiments, fluorescence intensities of each cell were obtained by setting a ROI according to the cell shape, and subtracting background values. Relative fluorescence intensities were calculated by setting the average intensity of the first three images (before PDGF stimulation) to 1.

**Statistics and reproducibility**. No statistical method was used to predetermine sample sizes. Samples were not randomized. The investigators were not blinded to allocation during experiments or outcome assessment. Sample sizes and statistical tests for each experiment are denoted in the figure legends. Dimedone-PLA images shown in each panel represent one biological replicate. Each dimedone-PLA experiment was performed at least twice (biological replicates), with quantification of six fields of images (technical replicates), except Fig. 3d (15 cells), per condition of the experiment. The absolute number of PLA puncta per cell in the same culture condition varies between biological replicates, presumably because of subtle differences in experimental conditions, but the fold-changes in number of PLA puncta between the various conditions were reproducible. Therefore, each graph for PLA quantification shows the result of one of the biological replicates. Each immunoblot was performed four times (biological replicates) for quantification, or at least twice (biological replicates) for expression checks. Each dimedone-PLA/ marker co-staining experiment or conventional immunostaining experiment was performed at least twice, and representative images from one of the biological replicates are shown in each panel. Pixel-based image analysis was performed with 10 (Fig. 5a) or 20 (Supplementary Fig. 8a) ROIs from ten cells per condition from one of the biological replicates, and object-based image analysis was performed with 50 cells per condition. The results of live cell imaging are from independent biological replicates. Statistical analysis was performed by using the two-tailed Welch's *t*-test (Figs. 1b and 5a, Supplementary Figs. 4c, 7c, 8a, 8c), unpaired/paired two-tailed *t*-test (Fig. 6c, Supplementary Figs. 4c, 7c, 8a, 9c), or ANOVA with Bonferroni/Dunn's post-hoc test (Figs. 1c, 2, 3b, 3d, 4a, 4b, 4c, Supplementary

Figs. 1d, 1e, 1g, 2a, 2b, 2c, 2e, 3a, 5a, 6c, 6d) by using StatView software, where appropriate. Precise *P*-values can be found in the figures.

**Code availability**. Computer codes for image analyses are available from the corresponding author upon request.

**Data availability**. Source data for figures are available from the corresponding author upon request.

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

## Acknowledgements

We thank Drs T. Kirchhausen (Harvard Medical School), N. Patel, H.W. Pauls (UHN Research, Toronto, Canada), and N.K. Tonks (Cold Spring Harbor Laboratory) for materials, Ms. X. Wang and Dr. R.S. Banh (Neel lab) for plasmids. We also thank Dr J. Jonkman (UHN Research, Toronto, Canada) for technical instruction and Dr M. Philips (NYU Medical Center) for helpful comments and discussion. This work

was supported by NIH R37 CA49132 (to B.G.N.), and by grants from the Deutsche Forschungsgemeinschaft (DFG) SFB 974 (to P.I.H.B. and R.S.), SFB815/TP1 and SCHR1241/1-1 (to K.S.). B.G.N. was also a Canada Research Chair, Tier 1, and work in his Toronto lab was partially supported by the Princess Margaret Cancer Foundation. R.T. was supported by a Postdoctoral Fellowship for Research Abroad of the Japan Society for the Promotion of Science (JSPS). Microscopy and flow cytometry experiments were supported by the Microscopy and Flow Cytometry and Cell Sorting Cores, respectively, of the Perlmutter Cancer Center, which are supported by NCI Cancer Center Support Grant P30 CA016887.

## Author contributions

R.T., P.I.H.B., and B.G.N. designed the research. R.T. performed experiments with the help of R.S., R.T., and J.H. analyzed the data. K.S. provided Nox1,2,4-triple KO MDFs. P.I.H.B. and R.S. provided HyPer3 expression vectors. R.T. and B.G.N. wrote the paper with advice from K.S. and P.I.H.B. B.G.N. provided overall supervision for the research.

## Additional information

**Competing interests:** The authors declare no competing financial interests.

