## [Peer Review File · Nature Communications]

Reviewers' Comments:

Reviewer #1 (Remarks to the Author):

In this manuscript, the authors present an imaging method to visualize oxidation of the protein tyrosine phosphatase SHP2 in response to PDGF stimulation. The method depends upon chemical modification of the oxidized SHP2 using dimedone and visualization with antibodies to dimedone and SHP2 using a proximity ligation assay. They suggest that oxidation of SHP2 is mediated by NOX enzymes, and occurs in “redoxosomes” in close proximity to R AB5+ endosomes. They propose that their approach may be adapted to study the oxidation of all members of the PTP family.

I think this is a potentially exciting contribution that should be of broad appeal to the signal transduction community, however, I think it would benefit from some additional clarification.

The entire approach depends upon the specific labeling of protein sulfenic acids by dimedone. Therefore, I think the authors should provide some evidence to support that specificity. Of the references cited, #16 simply states that “our labs and other groups have developed a series of reagents, based on dimedone, that specifically alkylate and, therefore, trap cysteine sulfenic acids”, but does not provide evidence of that specificity. Reference 17 describes antibodies as “sulfenic acid-specific” because they react with dimedone modified proteins. If this is really to be a general method then I think it is important to provide further validation of this critical step.

One specific problem is that the authors report on line 97-98 that “oxidized PTP1B can also be detected in H₂O₂-treated Swiss 3T3 fibroblasts by dimedone-PLA...” In contrast, Salmeen et al reported in Nature (2003) 423: p769 that following oxidation with H₂O₂, the sulfenic acid form was not detected, instead it was rapidly converted to a sulfenyl amide species. As cited by these authors, reference 38, the sulfenyl amide form has also been detected in cells. Often these chemical alkylations are slow, begging the question how is PTP1B detected by this method if dimedone is specific for cysteine sulfenic acids? Also, how much SHP2 is actually in the sulfenic acid form – reports from Rudolph’s lab [Chen et al (2009) Biochemistry 48: 1399-1409], for example, highlight a “back door cysteine” that may lead also to rapid conversion of the sulfenic acid to an S-S bond.

To what extent does global reactivity to dimedone change upon PDGF stimulation compared to H₂O₂? What % of the dimedone-labeled pool of proteins is SHP2?

What is the stoichiometry of oxidation of SHP2 in these studies? Previous papers, including from the Neel lab, have highlighted the difficulty in measuring the level of oxidation that occurs in response to a physiological stimulus compared to treatment with H₂O₂. Nevertheless, the data in Figure 1 suggest that PDGF and H₂O₂ are equally effective in generating the dimedone-PLA signal. In addition, the authors indicate on line 165 that, unlike following PDGF stimulation, ox-SHP2 signals do not colocalize with RAB5 in H₂O₂-treated cells. If “H₂O₂-evoked SHP2 oxidation .. can occur randomly in the cytoplasm”, wouldn’t it be expected that the extent of that oxidation would be greater than in response to PDGF? How does the stoichiometry of SHP2 oxidation compare between the two stimuli?

The manuscript is presented in a rather dense form that makes it difficult to extract the crucial points. It would be helpful if the authors could provide further clarification of their “redoxosome” model. In lines 26-27, they describe redoxosomes as “a specialized endosomal compartment”. The authors highlight (line 160) that “SHP2 binds to, and is endocytosed with, PDGFR β ” and (line 219-220) that ox-SHP2 signals were closely associated with NOX1 and NOX4, but (line 228) that “neither NOX1 nor NOX4 were visualized at the plasma membrane”. Endocytic vesicles are formed from the plasma membrane – what are their thoughts on how the NOX enzymes are incorporated into redoxosomes if they are not at the plasma membrane? Also, on line 236, they state that “SHP2 oxidation occurs on or close to RAB5+ endosomes” – what does “or close to” mean with respect to their redoxosome model? Also, if ox-SHP2 only occurs in endosomes, what are their thoughts on how the signal to trigger this comes from PDGF?

On line 301-303, the authors indicate that “treatment with the allosteric SHP2 inhibitor SHP099 restored PDGFR β tyrosine phosphorylation in TKO-MDFs, while having almost no effect on WT-MDFs”. This suggests that only receptor-bound SHP2 is important for regulation of MAPK signaling and that this pool is quantitatively oxidized and inactivated. I think it would be helpful if the authors could comment on their thoughts regarding the mechanism behind this extreme specificity for SHP2. Other groups have suggested a role for several PTPs in regulating PDGFR signaling, as well as other sources of ROS (such as mitochondria) – see for example, Frijhoff et al (2014) *Free Radical Biology & Medicine* 68: 268-277 – some comment on the differences between these various studies would be helpful.

Additional points.

On line 8, in the abstract – what are “sulfenylated cysteine residues”? Do the authors mean oxidation, or further modification of sulfenic acid?

On line 23, the authors state that “ROS are required for at least some signal transduction events”. Is “required” correct? The description of fine tuning of signaling (lines 46-47) seems more

appropriate.

On line 100, they state that “PDGF did not increase the number of puncta for oxidized PTP1B”. What happened with insulin stimulation? I think it is important to understand exactly how dimedone is labeling PTP1B.

On line 115, they state that “ox-SHP2 levels peaked around 5-10 mins” – how quickly is PDGF receptor endocytosed?

The authors seem to use “MDFs” and “MEFs” somewhat interchangeably – please clarify.

The data are admirably thorough and comprehensive, however, the images presented are often extremely small and consequently very difficult to view. If the size of the images could be increased it would greatly benefit the study, otherwise I think it would be of help to focus on a smaller number of representative images and use Supplementary Data to a greater extent.

Reviewer #2 (Remarks to the Author):

This is an important paper which demonstrates that PDGF can invoke transient oxidation of SHP2 at specific site within cells. The technique and results presented here will serve as a model to the redox community. I have a few additional experiments that would bolster the conclusion.

The authors should use better antioxidants than NAC. A key question is whether H₂O₂ is necessary for the oxidation. Antioxidants that diminish peroxide formation should be utilized. It could be that some other ROS is responsible for PDGF oxidation of SHP2.

Rhee and colleagues propose that peroxiredoxins must be inactivated either by phosphorylation or oxidation in order to effectively oxidize any protein in signal transduction. Do they detect with their technique oxidation of peroxiredoxins upon PDGF stimulation? PMID:20178744 and PMID:22147704

Reviewer #1 (Remarks to the Author):

In this manuscript, the authors present an imaging method to visualize oxidation of the protein tyrosine phosphatase SHP2 in response to PDGF stimulation. The method depends upon chemical modification of the oxidized SHP2 using dimedone and visualization with antibodies to dimedone and SHP2 using a proximity ligation assay. They suggest that oxidation of SHP2 is mediated by NOX enzymes, and occurs in “redoxosomes” in close proximity to RAB5+ endosomes. They propose that their approach may be adapted to study the oxidation of all members of the PTP family.

I think this is a potentially exciting contribution that should be of broad appeal to the signal transduction community, however, I think it would benefit from some additional clarification.

We appreciate Reviewer #1's comment that our research is “a potentially exciting contribution that should be of broad appeal to the signal transduction community.” We have performed a number of additional experiments to address his/her concerns, and added some clarifying information to the revised manuscript. We hope that he/she finds these responses illuminating, and that they resolve his/her remaining concerns.

The entire approach depends upon the specific labeling of protein sulfenic acids by dimedone. Therefore, I think the authors should provide some evidence to support that specificity. Of the references cited, #16 simply states that “our labs and other groups have developed a series of reagents, based on dimedone, that specifically alkylate and, therefore, trap cysteine sulfenic acids”, but does not provide evidence of that specificity. Reference 17 describes antibodies as “sulfenic acid-specific” because they react with dimedone modified proteins. If this is really to be a general method then I think it is important to provide further validation of this critical step.

We thank the Reviewer for pointing out this deficiency in the manuscript. In the revised paper, we include an additional reference to Poole *et al.* (2005, *Bioconjugate Chem.*), which reports that dimedone is incapable of labeling thiols or disulfide bonds (in the bacterial protein AhpC)¹.

We also conducted additional experiments to investigate further the specificity/preference of dimedone for the sulfenic acid oxidation state in the

mammalian cellular context. We treated cells with increasing concentrations of H₂O₂, with our goal being to achieve concentrations high enough to hyper-oxidize protein thiols beyond the sulfenic acid state to the (biologically irreversible) sulfinic or sulfonic states. Indeed, treatment of cells with 50 mM H₂O₂ resulted in a significant decrease in the intensity of anti-dimedone-Cys antibody staining, supporting the preferential reactivity of dimedone for reversible thiol oxidation. These data and the description of this experiment are found in the modified version of our manuscript (Supplementary Fig. 1e, P.7, L.97-99).

We also examined the ability of dimedone to label differentially oxidized forms of PTP1B *in vitro*. Again, we find that, consistent with the earlier studies (and the *in vivo* experiments cited above), dimedone cannot label irreversibly oxidized PTP1B. These experiments also suggest (although do not absolute show) that dimedone likely prefers to label the sulfenic acid state of PTP1B (PTP1B-SOH), rather than its sulfenylamide (PTP1B-SN) form (Supplementary Fig. 1a-c, P.6, L.67-87 and P.21, L363-371); please see more detailed discussion below.

One specific problem is that the authors report on line 97-98 that “oxidized PTP1B can also be detected in H₂O₂-treated Swiss 3T3 fibroblasts by dimedone-PLA...” In contrast, Salmeen et al reported in Nature (2003) 423: p769 that following oxidation with H₂O₂, the sulfenic acid form was not detected, instead it was rapidly converted to a sulfenyl amide species. As cited by these authors, reference 38, the sulfenyl amide form has also been detected in cells. Often these chemical alkylations are slow, begging the question how is PTP1B detected by this method if dimedone is specific for cysteine sulfenic acids?

We thank the Reviewer for raising this important issue. We investigated this problem by *in vitro* dimedone-labeling experiments. Salmeen *et al.* (Nature, 2003) reported that when PTP1B is treated with increasing concentrations of H₂O₂, it undergoes time-dependent reversible, and ultimately irreversible, oxidation. Concomitant crystallographic and MS analysis showed that only the sulfenylamide state of PTP1B (PTP1B-SN), not the sulfenic acid (PTP1B-SOH) could be detected under these conditions, leading them to conclude that the latter form of the enzyme was evanescent (at least under their *in vitro* conditions)².

Using an analogous experimental design, we asked how reversible oxidation relates to dimedone reactivity *in vitro*. Similar to Salmeen *et al.*², we find an H₂O₂ dose-dependent decrease in the reversibility of PTP1B oxidation, as reflected by PTP

activity measurements. By contrast, consistent with the previous work of Huyer *et al.*³, pervanadate causes irreversible PTP1B oxidation (Supplementary Fig. 1a).

We then labelled purified PTP1B with dimedone in the presence of DTT or the same, low concentrations of H₂O₂. This experiment enabled us to follow dimedone labeling of PTP1B during oxidation. As shown in Supplementary Fig. 1b of the revised manuscript, incubation of PTP1B with dimedone in the presence of H₂O₂ resulted in PTP1B labeling in a manner dependent on H₂O₂ concentration, whereas no dimedonylation of PTP1B occurred in the presence of pervanadate. We interpret these data as indicating that dimedone can label reversibly, but not irreversibly, oxidized PTP1B, although in fairness, we cannot be certain if dimedone is reacting with the PTP1B-SOH or PTP1B-SN form of the enzyme. Next, we pre-incubated PTP1B with H₂O₂, and *then* labelled with dimedone. As shown in Supplementary Fig. 1c, pre-incubation diminishes dimedone-labeling of PTP1B even with 80 μM H₂O₂, which only causes reversible oxidation (Supplementary Fig. 1a). Given that the work of Salmeen *et al.* shows that oxidation of PTP1B results in conversion to the sulfenylamide *in vitro*², this result suggests that dimedone does not label the sulfenylamide form effectively, or at least as effectively as the sulfenic acid form.

These findings beg the question of which reversible form of oxidized PTP1B is labelled in the cellular context. Tonks' group used an scFv specific for a sulfenylamide-mimicking mutant of PTP1B to visualize PTP1B oxidation response to cell stimulation⁴. They showed clearly that (some fraction of) reversibly oxidized PTP1B can exist in the sulfenylamide form inside cells. However, their findings do not exclude the existence of at least some PTP1B-SOH form; our data suggest that dimedone traps this portion of the enzyme. A corollary of this conclusion is that dimedone-treatment at the same time as the fixation step that we use in our assay is capable of labelling this PTP1B-SOH species.

In summary, we conclude that dimedone labels only reversibly (i.e., not irreversibly) oxidized PTP1B, and most likely prefers the sulfenic acid over the sulfenylamide form. Although our results suggest that only the PTP-SOH form can be labelled by dimedone, stronger evidence would require experiments of the type reported by Salmeen *et al.*², studies that we hope the Reviewer would agree are beyond the scope of this manuscript. We discuss these issues more thoroughly in the revised text (see P.6, L.67-87 and P.21, L363-371).

Also, how much SHP2 is actually in the sulfenic acid form – reports from Rudolph's lab [Chen et al (2009) *Biochemistry* 48: 1399-1409], for example, highlight a “back door

cysteine” that may lead also to rapid conversion of the sulfenic acid to an S-S bond.

This question is similar to the one above (and similarly important). We showed previously, using our “redox proteomics,” method, that ~10% of SHP2 is reversibly oxidized (sulfenic acid and disulfide bonds) in response to 1 mM H₂O₂, whereas SHP2 oxidation is undetectable by this proteomic assay in PDGF-stimulated cells⁵. Therefore, whatever form of SHP2 is being targeted by dimedone in cells, it is a small fraction of total SHP2.

We attempted to address the “backdoor cysteine”⁶ issue by asking whether mutation of these cysteines alters dimedone reactivity in cells. We expressed *Ptpn11*^{C333S/C367S} in *Ptpn11*^{f/f} cells, evoked deletion of endogenous (WT) *Ptpn11*, and used our assay to monitor SHP2 oxidation in response to either PDGF or H₂O₂. The dimedone-PLA signal was significantly reduced (see figure below for Reviewers only). At first glance, this result would seem to suggest that dimedone labels the disulfide form of reversibly oxidized SHP2 (SHP2-S-S) instead of the SHP2-SOH form. However, the study cited above also argues that the S-S form protects the catalytic cysteine from hyperoxidation to the SO₂H/SO₃H states⁶. Also, as noted above, the work of Poole *et al.* argues that dimedone is poorly reactive against S-S bonds (at least in the bacterial protein AhpC). Based on our results with PTP1B (above), we strongly suspect that dimedone-PLA preferentially detects the transient SHP2-SOH form, but we cannot exclude the possibility that the S-S form is also detected. Regardless, our results indicate that only reversibly oxidized SHP2 is visualized.

(a) *Ptpn11*^{fl/fl} MEFs expressing WT SHP2 or SHP2^{C333S/C367S} were treated with or without 4OHT, as indicated. Lysates were immunoblotted with anti-SHP2 and anti-ERK2 antibodies. (b) 4OHT-treated *Ptpn11*^{fl/fl} MEFs expressing WT SHP2 or SHP2^{C333S/C367S} were serum-starved, stimulated with PDGF-BB (50 ng/ml) or H₂O₂ (1 mM) for 10 min, fixed in the presence of dimedone (5 mM) for 5 min, and subjected to dimedone-PLA (gray). Representative images are shown for each condition from one of 2

independent experiments. The graph shows the average number of PLA signals per cell (n = 6 images for each condition, 5-20 cells in an image), relative to unstimulated control cells (normalized to 1). Error bars represent SD. ***P<0.0001, ANOVA with Bonferroni/Dunn's post-hoc test. Scale bar: 50 μ m.

To what extent does global reactivity to dimedone change upon PDGF stimulation compared to H₂O₂?

We do not detect a significant increase of anti-dimedone-Cys immunostaining in PDGF-stimulated cells, compared to unstimulated cells, whereas H₂O₂ causes a slight but significant increase (Supplementary Fig. 1d). We therefore think that: 1) there are significant amounts of dimedone-labeled proteins in cells even without stimulation, suggesting basal protein-thiol oxidation under our culture conditions; 2) Unlike H₂O₂, PDGF stimulation does not cause a global increase in dimedone-labeling, which is consistent with spatially and quantitatively limited protein oxidation in response to growth factors; and 3) immunostaining with anti-dimedone-Cys antibodies does not produce enough signal over noise to detect a PDGF-evoked increase in total protein oxidation over the background protein oxidation in unstimulated cells. These issues are discussed on P.7, L.94-97 of the revised manuscript.

What % of the dimedone-labeled pool of proteins is SHP2?

To address this question, *Ptpn11^{ff}* MEFs were subjected to tamoxifen treatment (to delete the floxed allele) or left untreated, and then immunostained with anti-dimedone-Cys antibodies. There was no significant difference in the intensity of the overall anti-dimedone signal in cells with or without SHP2 (Supplementary Fig. 1g). This result suggests that SHP2 is not a major protein that becomes oxidized and dimedone-labeled in cells. Consistent with this conclusion, anti-dimedone-Cys antibody immunoblotting of lysates from dimedone-labeled cells shows multiple bands other than SHP2 (Supplementary Fig. 1f), supporting the idea that SHP2 is a minor dimedonylated species. These experiments are described and discussed on P.7, L.99-P.8, L.105.

Nevertheless, the data in Figure 1 suggest that PDGF and H₂O₂ are equally effective in generating the dimedone-PLA signal. In addition, the authors indicate on line 165 that, unlike following PDGF stimulation, ox-SHP2 signals do not colocalize with RAB5 in H₂O₂-treated cells. If "H₂O₂-evoked SHP2 oxidation .. can occur randomly in the cytoplasm", wouldn't it be expected that the extent of that oxidation would be greater

than in response to PDGF? How does the stoichiometry of SHP2 oxidation compare between the two stimuli?

We apologize, but we are not exactly sure what the Reviewer is asking here. We think that ox-SHP2, as detected by dimedone-PLA, does not colocalize with RAB5 because it is (as the Reviewer states) occurring randomly in the cytoplasm when cells are treated with H₂O₂. As noted above, as measured by our redox proteomics approach, SHP2 oxidation in response to PDGF stimulation is less than in response to the 1 mM dose of H₂O₂ used in the dimedone-PLA experiments in Figure 1. We believe that our MS assay is intrinsically more quantitatively reliable than the dimedone-PLA approach presented here, and that significantly less SHP2 oxidation occurs in response to PDGF than 1mM H₂O₂. Whereas our new dimedone-PLA method is useful for visualizing the location of oxidation events, and gaining some assessment of relative amounts of oxidation, we doubt that it is linear. It should not be used for rigorous, quantitative assessments of oxidation events.

The manuscript is presented in a rather dense form that makes it difficult to extract the crucial points. It would be helpful if the authors could provide further clarification of their “redoxosome” model. In lines 26-27, they describe redoxosomes as “a specialized endosomal compartment”. The authors highlight (line 160) that “SHP2 binds to, and is endocytosed with, PDGFRβ” and (line 219-220) that ox-SHP2 signals were closely associated with NOX1 and NOX4, but (line 228) that “neither NOX1 nor NOX4 were visualized at the plasma membrane”. Endocytic vesicles are formed from the plasma membrane – what are their thoughts on how the NOX enzymes are incorporated into redoxosomes if they are not at the plasma membrane?

We are sorry that the Reviewer found the manuscript dense and the redoxosome concept difficult to follow. We were attempting to keep the word count low, and might have erred on the side of concision. In the revision, we tried to make the main concepts more clear (P.21, L.373-P.22, L.382). We also have added a schematic of our current model as Fig. 6d.

Although we observe NOX1 and NOX4 in vesicular structures, we cannot exclude the possibility that they also are found at the plasma membrane. Immunofluorescence staining highlights molecules that are concentrated in structures such as endosomes, whereas at the plasma membrane, the signal density could be more sparse and hard to see above the background. Notably, we did observe PDGF-evoked

co-localization of PDGFR β and NOX proteins (Fig. 5a). Therefore, we think that there are two possibilities: (1) NOX proteins also are present at fairly low (undetectable) levels on the plasma membrane, and are endocytosed with the PDGFR to form redoxosomes; or (2) NOX proteins are on cytoplasmic vesicular structures and these vesicles are, by an unknown mechanism, recruited to, and fused with, endocytic vesicles containing PDGFR. We hope that the Reviewer will agree that distinguishing between these possibilities is beyond the scope of this manuscript. We added this discussion in the revised manuscript (P.22,L.382-389).

Also, on line 236, they state that “SHP2 oxidation occurs on or close to RAB5+ endosomes” – what does “or close to” mean with respect to their redoxosome model?

We were trying to be very cautious in our conclusions. We think that it is highly likely that SHP2 is bound to PDGFR on redoxosomes when it gets oxidized (i.e., it is “on” redoxosomes). But the spatial resolution of the PLA signals by light microscopy and object-based image analysis does not permit such a conclusion to be made with certainty (i.e., oxidation could occur anywhere within the diffusion limit of H₂O₂ from redoxosomes). Therefore, we think it is more accurate to say “on or close to” RAB5+ endosomes. Descriptions related to this issue can be found on P.17, L.284-288.

Also, if ox-SHP2 only occurs in endosomes, what are their thoughts on how the signal to trigger this comes from PDGF?

PDGFR β is endocytosed following PDGF stimulation, and numerous previous studies have shown that receptor tyrosine kinases, including PDGFR β , continue to signal while on endosomes⁷⁻¹⁰. Presumably, one of these signals triggers H₂O₂ production. It has been reported that the PI3K-RAC axis is required for growth factor-evoked ROS generation¹¹⁻¹³. Consistent with this model, we observed that PI3K inhibitors suppressed PDGF-evoked SHP2 oxidation (new Supplementary Fig. 6d, P.14, L.239-241). Surprisingly, however, preliminary experiments using dominant negative RAC argue against a role for this small G protein in SHP2 oxidation. Further experiments will be required to resolve the detailed mechanism, but we would argue (and hope the Reviewer agrees) that such work is beyond the scope of this paper.

On line 301-303, the authors indicate that “treatment with the allosteric SHP2 inhibitor SHP099 restored PDGFR β tyrosine phosphorylation in TKO-MDFs, while having

almost no effect on WT-MDFs". This suggests that only receptor-bound SHP2 is important for regulation of MAPK signaling and that this pool is quantitatively oxidized and inactivated. I think it would be helpful if the authors could comment on their thoughts regarding the mechanism behind this extreme specificity for SHP2. Other groups have suggested a role for several PTPs in regulating PDGFR signaling, as well as other sources of ROS (such as mitochondria) – see for example, Frijhoff et al (2014) *Free Radical Biology & Medicine* 68: 268-277 – some comment on the differences between these various studies would be helpful.

Our description must have been unclear, and we apologize. We are not saying that PDGFR dephosphorylation is catalyzed *only* by SHP2. We observed an ~40% decrease in PDGFR tyrosyl phosphorylation in NOX1,2,4-TKO MDFs. SHP099 treatment *partially* restores phosphorylation of these sites, leaving open the possibility that other ROS-targeted PTPs also help to regulate PDGFR tyrosyl phosphorylation. We modified the description on P.20, L.344, to note SHP099 *partially* restores decreased PDGFR β phosphorylation. Also, we examined phosphorylation at a single time after stimulation, and it is quite possible, if not likely, that other PTPs contribute either before or after PDGFR stimulation (e.g., RPTPs most likely restrain auto-activation).

In addition, our *Ptpn11* knockout experiments, as well as SHP2 inhibitor studies, indicate that SHP2 is necessary for downstream ERK pathway activation even at the time that SHP2 is oxidized. These findings suggest that catalytically active SHP2 functions as a positive regulator of the ERK pathway *before* oxidation or there is an activated SHP2 pool distant from the redoxosomes that mediates ERK activation (P.24, L.416-417).

Frijhoff *et al.*¹⁴ reported decreased PDGFR phosphorylation in p66SHC-knockout or knockdown cells, and showed overall depletion of oxidized PTPs in p66SHC-knockout MEFs treated with PDGF as well as untreated MEFs. They further showed that, when assessed by SHP2 phosphatase activity in the presence or absence of reducing agents, 40% of total SHP2 is oxidized in unstimulated wild-type cells and 70% of SHP2 is oxidized in PDGF-stimulated wild-type cells, whereas 20% of SHP2 is oxidized in unstimulated p66SHC-KO cells and 50% of SHP2 is oxidized in PDGF-stimulated p66SHC-KO cells. On the other hand, however, they also showed that PDGF-dependent SHP2 oxidation, as detected by cysteinyl-labeling, was almost totally abolished in the KO cells. With all due respect to these investigators, it is quite difficult to use PTP activity assays to assess PTP oxidation, given the ease of gratuitous oxidation that can occur with such an experimental design. Regardless, even if we

accept their conclusions at face value, their experiments indicate the presence of *both* p66SHC/mitochondria-dependent and -independent ROS production by PDGF, and suggest that ROS from mitochondria via p66SHC contributes to basal oxidation rather than to the magnitude of PDGF-induced increase of SHP2 oxidation. We have discussed these issues further in the revised text (P.24. L.424-430).

Additional points.

On line 8, in the abstract – what are “sulfenylated cysteine residues”? Do the authors mean oxidation, or further modification of sulfenic acid?

We thank the Reviewer for pointing out this confusing nomenclature. We have modified the description to “cysteine residues in the sulfenic acid state” (P.2, L.8).

On line 23, the authors state that “ROS are required for at least some signal transduction events”. Is “required” correct? The description of fine tuning of signaling (lines 46-47) seems more appropriate.

Sundaresan *et al.* showed suppressed PDGF signaling in the absence of ROS¹⁵, so we feel that “required” is not inaccurate. However, to avoid any confusion, we have modified this statement to “...ROS are required for precise regulation of at least some signal transduction events” (P.3, L.21-22).

On line 100, they state that “PDGF did not increase the number of puncta for oxidized PTP1B”. What happened with insulin stimulation? I think it is important to understand exactly how dimedone is labeling PTP1B.

Because Swiss3T3 cells, which were the predominant cell type used in this study, do not respond strongly to insulin, we did not show this result. In response to the Reviewer, however, we addressed this issue by using another cell system. Specifically, we engineered human HepG2 cells to express *PTPNI* shRNA to knockdown endogenous PTP1B, and then expressed either mouse wild-type PTP1B or mutant PTP1B in which the catalytic cysteine (C215) residue is substituted by a serine residue (PTP1B^{C215S}). As shown in the new Fig. 2 and described on P.9, L.135-140, we observed enhancement of dimedone-PLA signals upon insulin-treatment in wild-type PTP1B-expressing, but not PTP1B^{C215S}-expressing, cells. These results demonstrate that

our method can detect insulin-evoked PTP1B oxidation, and shows that PTP1B oxidation occurs on (or at least requires intact) catalytic cysteine. We thank the Reviewer for this comment, which allowed us to demonstrate further the potential generality of our method.

On line 115, they state that “ox-SHP2 levels peaked around 5-10 mins” – how quickly is PDGF receptor endocytosed?

We immunostained PDGFR β in starved and stimulated Swiss 3T3 cells, and observed a detectable decrease of surface PDGFR β after 2.5 min after PDGF addition (Supplementary Fig. 5b). These findings (discussed on P.10, L.158-161 of the revised manuscript) support our conclusion that endocytosis of PDGFR, PDGF-dependent ROS production at RAB5+ vesicles, and oxidation of SHP2 occur at similar times after stimulation.

The authors seem to use “MDFs” and “MEFs” somewhat interchangeably – please clarify.

We apologize to the Reviewer for this confusion. “MDFs,” as defined in the text, refer to mouse DERMAL fibroblasts. The NOX-deficient and control WT cells are “MDFs”. “MEFs” refer to mouse embryo fibroblasts. The *Ptpn11*^{fl/fl} cells are immortalized MEFs. Unfortunately, we must retain this nomenclature for accuracy of the description of these different types of fibroblasts, but we have made certain that we are careful in the use of these abbreviations in the text.

The data are admirably thorough and comprehensive, however, the images presented are often extremely small and consequently very difficult to view. If the size of the images could be increased it would greatly benefit the study, otherwise I think it would be of help to focus on a smaller number of representative images and use Supplementary Data to a greater extent.

We thank the Reviewer for his/her advice. We have the enlarged images in the revised manuscript.

Reviewer #2 (Remarks to the Author):

This is an important paper which demonstrates that PDGF can invoke transient oxidation of SHP2 at specific site within cells. The technique and results presented here will serve as a model to the redox community. I have a few additional experiments that would bolster the conclusion.

We thank the Reviewer for his/her positive comments on our manuscript. We hope that the additional results described below, as well as in response to Reviewer 1, can bolster our conclusions.

The authors should use better antioxidants than NAC. A key question is whether H₂O₂ is necessary for the oxidation. Antioxidants that diminish peroxide formation should be utilized. It could be that some other ROS is responsible for PDGF oxidation of SHP2.

As the Reviewer notes, our manuscript did not specify which type of ROS is responsible for PDGF-evoked SHP2 oxidation. To answer this question, we expressed a catalase mutant lacking a peroxisome targeting sequence in Swiss 3T3 cells. As shown in Fig. 4a and discussed on P.14, L.231-234, expression of cytoplasmic catalase suppressed PDGF-evoked enhancement of dimedone-PLA signals, strongly arguing for the involvement of H₂O₂ in this process.

Rhee and colleagues propose that peroxiredoxins must be inactivated either by phosphorylation or oxidation in order to effectively oxidize any protein in signal transduction. Do they detect with their technique oxidation of peroxiredoxins upon PDGF stimulation? PMID:20178744 and PMID:22147704

We thank Reviewer for this question, which, like the insulin experiments suggested by Reviewer 1, gave us the chance to try to further explore the generality of our method. We tried several commercially available antibodies to PRDX1 (Santa Cruz sc-59657) and PRDX2 (Santa Cruz sc-515428 and Proteintech 60202-1-Ig), respectively. However, with these antibodies, dimedone-PLA either did not generate any signal (anti-PRDX1 antibody, sc-59657; anti-PRDX2 antibody, sc-515428) or generated dimedone-PLA signals even in *PRDX2* knockdown cells generated by siRNA (anti-PRDX2 antibody, 60202-1-Ig). Therefore, we unfortunately are unable to address the issue raised by the Reviewer with available reagents.

References

1. Poole, L. B., Zeng, B. B., Knaggs, S. A., Yakubu, M. & King, S. B. Synthesis of chemical probes to map sulfenic acid modifications on proteins. *Bioconjug Chem.* **16**, 1624-1628 (2005).
2. Salmeen, A. *et al.* Redox regulation of protein tyrosine phosphatase 1B involves a sulphenyl-amide intermediate. *Nature* **423**, 769-773 (2003).
3. Huyer, G. *et al.* Mechanism of inhibition of protein-tyrosine phosphatases by vanadate and pervanadate. *J. Biol. Chem.* **272**, 843-851 (1997).
4. Haque, A., Andersen, J. N., Salmeen, A., Barford, D. & Tonks, N. K. Conformation-sensing antibodies stabilize the oxidized form of PTP1B and inhibit its phosphatase activity. *Cell* **147**, 185-198 (2011).
5. Karisch, R. *et al.* Global proteomic assessment of the classical protein-tyrosine phosphatome and "Redoxome". *Cell* **146**, 826-840 (2011).
6. Chen, C. Y., Willard, D. & Rudolph, J. Redox regulation of SH2-domain-containing protein tyrosine phosphatases by two backdoor cysteines. *Biochemistry* **48**, 1399-1409 (2009).
7. Wang, Y., Pennock, S. D., Chen, X., Kazlauskas, A. & Wang, Z. Platelet-derived growth factor receptor-mediated signal transduction from endosomes. *J. Biol. Chem.* **279**, 8038-8046 (2004).
8. Vieira, A. V., Lamaze, C. & Schmid, S. L. Control of EGF receptor signaling by clathrin-mediated endocytosis. *Science* **274**, 2086-2089 (1996).
9. Burke, P., Schooler, K. & Wiley, H. S. Regulation of epidermal growth factor receptor signaling by endocytosis and intracellular trafficking. *Mol. Biol. Cell.* **12**, 1897-1910 (2001).
10. Wang, Y., Pennock, S., Chen, X. & Wang, Z. Endosomal signaling of epidermal growth factor receptor stimulates signal transduction pathways leading to cell survival.

Mol. Cell. Biol. **22**, 7279-7290 (2002).

11. Sundaresan, M. *et al.* Regulation of reactive-oxygen-species generation in fibroblasts by Rac1. *Biochem J.* **318**, 379-382 (1996)

12. Bae, Y. S. *et al.* Platelet-derived growth factor-induced H₂O₂ production requires the activation of phosphatidylinositol 3-kinase. *J. Biol. Chem.* **275**, 10527-10531 (2000).

13. Park, H. S. *et al.* Sequential activation of phosphatidylinositol 3-kinase, \square Pix, Rac1, and Nox1 in growth factor-induced production of H₂O₂. *Mol. Cell. Biol.* **24**, 4384-4394 (2004).

14. Frijhoff, J. *et al.* The mitochondrial reactive oxygen species regulator p66Shc controls PDGF-induced signaling and migration through protein tyrosine phosphatase oxidation. *Free Radic. Biol. Med.* **68**, 268-277 (2014).

15. Sundaresan, M., Yu, Z. X., Ferrans, V. J., Irani, K. & Finkel, T. Requirement for generation of H₂O₂ for platelet-derived growth factor signal transduction. *Science* **270**, 296-299 (1995).

16. Woo, H. A. *et al.* Inactivation of peroxiredoxin I by phosphorylation allows localized H₂O₂ accumulation for cell signaling. *Cell* **140**, 517-528 (2010).

Reviewers' Comments:

Reviewer #1 (Remarks to the Author):

I have reviewed the revised manuscript and recommend that it be accepted for publication at this stage.

Reviewer #2 (Remarks to the Author):

I am satisfied.